# Majorana Diagrammatics for Quantum Spin-½ Models

**Thibault Noblet[1]⋆, Laura Messio[1]⊗ and Riccardo Rossi[2,1]†**

**1** Sorbonne Université, CNRS, Laboratoire de Physique Théorique de la Matière
Condensée, LPTMC, F-75005 Paris, France
**2** Institute of Physics, École Polytechnique Fédérale de Lausanne (EPFL), CH-1015
Lausanne, Switzerland

⋆ thibault.noblet@sorbonne-universite.fr ,    ⊗ laura.messio@sorbonne-universite.fr ,
† riccardo.rossi@epfl.ch

## Abstract

A diagrammatic formalism for lattices of spin-½ is developed. It is based on
an unconstrained mapping between spin and Majorana operators. This allows
the use of standard tools of diagrammatic quantum many-body theory without
requiring projections. We derive, in particular, the Feynman rules for the
expansion around a color-preserving mean-field theory. We then present the
numerical results obtained by computing the corrections up to second order for
the Heisenberg model in one and two dimensions, showing that perturbative
corrections are not only numerically important, but also qualitatively improve
the results of mean-field theory. These results pave the way for the use of
Majorana diagrammatic tools in theoretical and numerical studies of quantum
spin systems.

# 1  Introduction

Systems of spin-$\frac{1}{2}$ on a lattice have given rise to a rich variety of phases and phenomena. In particular, frustration can prevent conventional magnetic order, resulting in Quantum Spin Liquids (QSLs) [1–3], without any spontaneous symmetry breaking at $T = 0$. The Kitaev model [4] is an emblematic model presenting both gapped and ungapped QSL

phases, with the remarquable specificity to have a known exact solution. However, many frustrated spin models lack such exact solutions and require approximate analytical or numerical approaches.

A common theoretical strategy to investigate QSLs is to employ parton representations, where each spin operator is expressed in terms of more tractable fermionic [5–7] or bosonic [8–10] degrees of freedom. However, a major drawback of standard parton approaches is that they enlarge the Hilbert space, introducing unphysical states that do not correspond to any real spin configuration. While it is possible to deal with this constraint with Variational Monte Carlo [11–13], it is challenging to impose the restriction to the physical part of the Hilbert space in a field-theory framework, impeding the systematic calculation of corrections to the mean-field theory.

The Popov-Fedotov trick (PFT) [14, 15] allows to exactly perform the projection in the fermionic parton representation by introducing a complex chemical potential that, at finite temperature, eliminates the contribution of unphysical states. The PFT allows the use of field-theory techniques to quantum spin systems and to compute physical quantities in a systematic way with Diagrammatic Monte Carlo [16–22]. However, the PFT requires the use of a non-Hermitian Hamiltonian, it works only at finite temperature, and unphysical states still appear in intermediate calculations at all expansion orders, as they are eliminated only after the whole perturbative series is summed.

The $SO(3)$ Majorana fermion representation of spin-$\frac{1}{2}$ operators [23–30], closely related to the drone-fermion [31–33] and Grassmann-variable [34–36] approaches, offers an elegant route to avoid these complications by providing a constraint-free mapping. It expresses each spin operator in terms of three Majorana fermions and exactly reproduces the $SU(2)$ spin algebra without requiring any constraint, in contrast with the four-Majorana representation introduced to solve the Kitaev honeycomb model [4, 30]. The mapping creates copies of the original spin Hilbert space, which manifests itself in a $\mathbb{Z}_2$ gauge redundancy, and no unphysical state is introduced. We note that this representation is used in the recently-proposed Pseudo-Majorana Functional Renormalization Group method [37–41].

In this paper, we introduce a general diagrammatic formalism for quantum spin-$\frac{1}{2}$ models using the $SO(3)$ Majorana representation. Specifically, we consider a Majorana mean-field Ansatz as a perturbative starting point, and we systematically compute the corrections due to the interaction. After having discussed the properties of the representation and its gauge redundancy, we derive the Feynman diagram rules for a color-preserving mean-field expansion, which are used to write down explicitly the correction to the spin susceptibility up to second order. We benchmark our approach on the antiferromagnetic Heisenberg model in one and two dimensions. The diagrammatic corrections are found to be both quantitatively and qualitatively important as they restore key features that the mean-field theory misses. Our results show that the Majorana diagrammatic approach we develop can systematically correct even a qualitatively-wrong mean-field theory result, and it could provide an additional theoretical and numerical tool for the study of quantum spin systems.

## 2 Formalism

The main goal of this work is to develop a systematic diagrammatic formalism for quantum spin-$\frac{1}{2}$ models. To achieve this, as mentioned, we use a constraint-free Majorana-fermion representation of the spin-$\frac{1}{2}$ degrees of freedom, the $SO(3)$ Majorana representation, as detailed in Sec. 2.1. In Sec. 2.2 we show how to retrieve physical spin observables from the Majorana-fermion representation without projections, and we discuss the gauge choice at

the interacting level. In Sec. 2.3, in the context of the $XYZ$ model, we introduce the color-preserving Majorana mean-field theory that we use as perturbative starting point. Finally, Sec. 2.4 is dedicated to the derivation of the Feynman rules of the expansion around the color-preserving mean-field theory, and we give explicit expressions for spin-susceptibility Feynman diagrams up to second order.

## 2.1 $SO(3)$ Majorana Representation for spin-$\frac{1}{2}$ particles

This section is dedicated to the discussion of the properties of the $SO(3)$ Majorana representation for spin-$\frac{1}{2}$, used in this work. As already mentioned, its advantage over other representations such as the Kitaev's, which uses four Majorana fermions per spin, or the Abrikosov-fermion one, which uses two fermions per spin, is that no constraint needs to be imposed, as the Hilbert space consists in multiple copies of the physical spin-$\frac{1}{2}$ Hilbert space, which produce a $\mathbb{Z}_2$ gauge redundancy. Note that the Jordan-Wigner transformation is another example of constraint-less mapping for spin-$\frac{1}{2}$, but with the disadvantage of being non-local.

### 2.1.1 Definition

The $SO(3)$ Majorana representation for spin-$\frac{1}{2}$ particles [23,30] is defined as:

$$\hat{\mathcal{S}}_j^\alpha := -\frac{i}{4} \sum_{\gamma,\delta \in \{x,y,z\}} \epsilon_{\alpha\gamma\delta} \, \hat{\rho}_j^\gamma \, \hat{\rho}_j^\delta, \qquad j \in \mathcal{L}, \quad \alpha \in \{x,y,z\}, \tag{1}$$

where Latin letters such as $j \in \mathcal{L}$ denote a site-index on a lattice $\mathcal{L}$, Greek letters such as $\alpha \in \{x,y,z\}$ are used for three-dimensional components, $\hat{\mathcal{S}}_j^\alpha$ is a Majorana spin-$\frac{1}{2}$ operator, $\epsilon_{\alpha\gamma\delta}$ is the Levi-Civita tensor, and $\hat{\rho}_j^\alpha$ is a Majorana-fermion operator, which is Hermitian and satisfies the following anticommutation relations

$$\{\hat{\rho}_j^\gamma, \hat{\rho}_k^\delta\} = 2 \, \delta_{jk} \, \delta_{\gamma\delta}. \tag{2}$$

We remark that if we introduce the vectors of operators $\hat{\boldsymbol{\mathcal{S}}}_j = (\hat{\mathcal{S}}_j^x, \hat{\mathcal{S}}_j^y, \hat{\mathcal{S}}_j^z)^T$ and $\hat{\boldsymbol{\rho}}_j = (\hat{\rho}_j^x, \hat{\rho}_j^y, \hat{\rho}_j^z)^T$, Eq. (1) simply writes as

$$\hat{\boldsymbol{\mathcal{S}}}_j := -\frac{i}{4} \, \hat{\boldsymbol{\rho}}_j \times \hat{\boldsymbol{\rho}}_j, \qquad j \in \mathcal{L}. \tag{3}$$

One can verify that the Majorana spin-$\frac{1}{2}$ operators $\hat{\mathcal{S}}_j^\alpha$ defined in Eq. (1) are Hermitian and satisfy all the properties of spin-$\frac{1}{2}$ operators:

$$(\hat{\mathcal{S}}_j^\alpha)^2 = \frac{1}{4}, \qquad [\hat{\mathcal{S}}_j^\alpha, \hat{\mathcal{S}}_k^\gamma] = i\delta_{j,k} \sum_\delta \epsilon_{\alpha\gamma\delta} \, \hat{\mathcal{S}}_j^\delta. \tag{4}$$

We emphasize that the $SO(3)$ representation employed here differs from Kitaev's construction [4,30] in two key aspects: (i) it requires only three Majorana fermions, and (ii) it imposes the spin-$\frac{1}{2}$ algebra without any constraint.

### 2.1.2 Spin-$\frac{1}{2}$, Majorana, and copy Hilbert spaces

In the following, we denote by $\hat{S}_j^\alpha$ a spin-$\frac{1}{2}$ operator acting on the physical spin Hilbert space $\mathcal{H}_{\text{spin}}$, and by $\hat{\mathcal{S}}_j^\alpha$ the Majorana spin-$\frac{1}{2}$ operator defined in Eq. (1), acting on the Majorana Hilbert space $\mathcal{H}_{\text{Maj}}$. We consider an even number of lattice sites, $|\mathcal{L}| = 2N$, with a spin-$\frac{1}{2}$ physical degree of freedom per lattice site. The dimension of $\mathcal{H}_{\text{spin}}$ is then

$2^{2N}$. Two Majorana fermion operators can be paired to form a complex fermion, giving a dimension of $\mathcal{H}_{\mathrm{Maj}}$ of $2^{3N}$.

As the Majorana spin-$\frac{1}{2}$ operators satisfy the spin-$\frac{1}{2}$ algebraic properties, $\mathcal{H}_{\mathrm{Maj}}$ consists of *copies* of $\mathcal{H}_{\mathrm{spin}}$

$$\mathcal{H}_{\mathrm{Maj}} = \mathcal{H}_{\mathrm{spin}} \otimes \mathcal{H}_{\mathrm{copy}}, \tag{5}$$

where $\mathcal{H}_{\mathrm{copy}}$ is a "copy" Hilbert space, of dimension $2^N$. The Majorana spin operators $\hat{\mathcal{S}}_j^\alpha$ write

$$\hat{\mathcal{S}}_j^\alpha = \hat{S}_j^\alpha \otimes \hat{\mathbb{1}}_{\mathrm{copy}}, \tag{6}$$

where $\hat{\mathbb{1}}_{\mathrm{copy}}$ is the identity operator acting on $\mathcal{H}_{\mathrm{copy}}$.

### 2.1.3 Spin rotation

The Majorana spin operators vector $\hat{\boldsymbol{\mathcal{S}}}_j = (\hat{\mathcal{S}}_j^x, \hat{\mathcal{S}}_j^y, \hat{\mathcal{S}}_j^z)^T$ and the Majorana operator vector $\hat{\boldsymbol{\rho}}_j = (\hat{\rho}_j^x, \hat{\rho}_j^y, \hat{\rho}_j^z)^T$ both transform as vectors under a local spin rotation. For $R \in SO(3)$,

$$\hat{\boldsymbol{\rho}}_j \mapsto R\,\hat{\boldsymbol{\rho}}_j \qquad \Rightarrow \qquad \hat{\boldsymbol{\mathcal{S}}}_j \mapsto R\,\hat{\boldsymbol{\mathcal{S}}}_j. \tag{7}$$

### 2.1.4 $\mathbb{Z}_2$ gauge invariance

We have seen that the Majorana representation is redundant: the dimension of $\mathcal{H}_{\mathrm{Maj}}$ is exponentially higher than the dimension of $\mathcal{H}_{\mathrm{spin}}$. This manifests itself in a $\mathbb{Z}_2$ gauge redundancy of the fermion-to-spin mapping, as we make explicit in this section.

For $\sigma_j \in \{-1, 1\}$, we define a $\mathbb{Z}_2$ gauge transformation $G$ by

$$\hat{\boldsymbol{\rho}}_j \overset{G}{\mapsto} \sigma_j\,\hat{\boldsymbol{\rho}}_j. \tag{8}$$

From Eq. (1), we get that the Majorana spin operators are invariant under this $\mathbb{Z}_2$ gauge transformation

$$\hat{\boldsymbol{\mathcal{S}}}_j \overset{G}{\mapsto} \hat{\boldsymbol{\mathcal{S}}}_j. \tag{9}$$

### 2.1.5 Majorana Triple Operators

Following Refs. [30, 42, 43], we introduce the Majorana triple operators

$$\hat{\tau}_j = -i\,\hat{\rho}_j^x\,\hat{\rho}_j^y\,\hat{\rho}_j^z = -\frac{i}{6}\,\hat{\boldsymbol{\rho}}_j \cdot \hat{\boldsymbol{\rho}}_j \times \hat{\boldsymbol{\rho}}_j, \tag{10}$$

for $j \in \mathcal{L}$. The Majorana triple operators are Hermitian, commute with the Majorana spin operators,

$$[\hat{\tau}_j, \hat{\mathcal{S}}_l^\alpha] = 0, \tag{11}$$

but do not commute with each other. Thus, the Majorana triple operators act trivially on $\mathcal{H}_{\mathrm{spin}}$, and, accordingly, we can define $\hat{t}_j : \mathcal{H}_{\mathrm{copy}} \to \mathcal{H}_{\mathrm{copy}}$ such that

$$\hat{\tau}_j = \hat{\mathbb{1}}_{\mathrm{spin}} \otimes \hat{t}_j. \tag{12}$$

### 2.1.6 Copy-pair operators and complete basis for the Majorana Hilbert space

We introduce the copy-pair operators $\hat{\mathcal{J}}_{jl}^z$ [30]

$$\hat{\mathcal{J}}_{jl}^z := -\frac{i}{2}\,\hat{\tau}_j\,\hat{\tau}_l, \tag{13}$$

which are Hermitian operators with eigenvalues $\pm\frac{1}{2}$. The copy-pair operators act non-trivially only on $\mathcal{H}_{\text{copy}}$, see Eq. (12). Under a gauge transformation $G$, they transform as

$$\hat{\mathcal{J}}_{jl}^z \xmapsto{G} \sigma_j\,\sigma_l\,\hat{\mathcal{J}}_{jl}^z. \tag{14}$$

Two copy-pair operators on four different sites $j$, $k$, $l$ and $m$ commute, $[\hat{\mathcal{J}}_{jk}^z, \hat{\mathcal{J}}_{lm}^z] = 0$.

As the lattice $\mathcal{L}$ has an even number of points equal to $2N$, we can consider a pairing of lattice sites $\mathcal{D} = \{(j_1, l_1), \ldots, (j_N, l_N)\}$, such that $\{j_1, \ldots, j_N, l_1, \ldots, l_N\} = \mathcal{L}$. The copy-pair operators built on these pairs commute with each other and, from Eq. (11), they also commute with the Majorana spin operators $\hat{\mathcal{S}}_j^\alpha$. Therefore, as $\mathcal{H}_{\text{copy}}$ has dimension $2^N$, $\hat{\mathcal{J}}_{jk}^z$ for $(j, k) \in \mathcal{D}$ and $\hat{\mathcal{S}}_l^z$ for $l \in \mathcal{L}$ provide a complete set of commuting observables for $\mathcal{H}_{\text{Maj}}$.

## 2.2 Mapping a spin-$\frac{1}{2}$ Hamiltonian to a Majorana Hamiltonian

In this section, we show how the expectaction value of physical observables at thermal equilibrium are obtained from Majorana-fermion expectation values without any projection. We also discuss what we refer to as interacting-level gauge choice: we are free to add an arbitrary function of the Majorana triple operators to the Majorana-mapped spin Hamiltonian without affecting physical results.

### 2.2.1 Mapping of spin-$\frac{1}{2}$ terms to Majorana spin operators

We start with a Hamiltonian $\hat{H} : \mathcal{H}_{\text{spin}} \to \mathcal{H}_{\text{spin}}$

$$\hat{H} = H(\hat{\boldsymbol{S}}), \tag{15}$$

where $H$ is a function of the $\hat{\boldsymbol{S}} = (\hat{S}_j^\alpha)_{j\in\mathcal{L},\, \alpha\in\{x,y,z\}}$ operators. We associate to $\hat{H}$ the Hamiltonian $\hat{\mathcal{H}} : \mathcal{H}_{\text{Maj}} \to \mathcal{H}_{\text{Maj}}$ obtained by directly substituting the spin-$\frac{1}{2}$ operators $\hat{S}_j^\alpha$ with the Majorana spin operators $\hat{\mathcal{S}}_j^\alpha$

$$\hat{\mathcal{H}} = H(\hat{\boldsymbol{\mathcal{S}}}), \tag{16}$$

where $\hat{\boldsymbol{\mathcal{S}}} = (\hat{\mathcal{S}}_j^\alpha)_{j\in\mathcal{L},\, \alpha\in\{x,y,z\}}$. Following the discussion of Sec. 2.1.2,

$$\hat{\mathcal{H}} = \hat{H} \otimes \hat{\mathbb{1}}_{\text{copy}}. \tag{17}$$

### 2.2.2 Copy Hamiltonian

We could also consider the more general Majorana Hamiltonian

$$\hat{\mathcal{H}}_{\text{Maj}} = \hat{\mathcal{H}} + \hat{\mathcal{K}}, \qquad \hat{\mathcal{K}} = K(\hat{\boldsymbol{\tau}}) \tag{18}$$

where $\hat{\boldsymbol{\tau}} = (\hat{\tau}_j)_{j\in\mathcal{L}}$ are the Majorana triple operators of Eq. (10), and $K(\hat{\boldsymbol{\tau}})$ is a function of the $\hat{\boldsymbol{\tau}}$ defining the copy Hamiltonian $\hat{\mathcal{K}}$. As the Majorana triple operators commute with the Majorana spin operators, they act non-trivially only on $\mathcal{H}_{\text{copy}}$

$$\hat{\mathcal{K}} = \hat{\mathbb{1}}_{\text{spin}} \otimes \hat{K}, \tag{19}$$

where $\hat{K}$ is an operator acting on $\mathcal{H}_{\text{copy}}$.

### 2.2.3    Thermal density matrix and expectation values

In this work, we focus on thermal properties at inverse temperature $\beta$, which is a dummy variable in most of the following equations. The partition function with the Majorana Hamiltonian $\hat{\mathcal{H}}_{\mathrm{Maj}}$, defined in Eq. (18) is

$$\mathcal{Z}_{\mathrm{Maj}} = \mathrm{Tr}_{\mathrm{Maj}}\, e^{-\beta \hat{\mathcal{H}}_{\mathrm{Maj}}} = \mathrm{Tr}_{\mathrm{Maj}}\, e^{-\beta(\hat{H}\otimes\hat{\mathbb{1}}_{\mathrm{copy}} + \hat{\mathbb{1}}_{\mathrm{spin}}\otimes\hat{K})} = Z\, Z_{\mathrm{copy}}, \tag{20}$$

where we have used Eq. (16) and Eq. (19), and where we have introduced the physical spin-$\frac{1}{2}$ partition function $Z$ and the partition function of the copy system $Z_{\mathrm{copy}}$

$$Z = \mathrm{Tr}_{\mathrm{spin}}\, e^{-\beta \hat{H}}, \qquad Z_{\mathrm{copy}} = \mathrm{Tr}_{\mathrm{copy}}\, e^{-\beta \hat{K}}. \tag{21}$$

The thermal density matrix $\hat{\eta}_\beta$ is the product of a spin and a copy density matrix

$$\hat{\eta}_\beta = \frac{e^{-\beta \hat{\mathcal{H}}_{\mathrm{Maj}}}}{\mathcal{Z}_{\mathrm{Maj}}} = \hat{\eta}_{\beta;\mathrm{spin}} \otimes \hat{\eta}_{\beta;\mathrm{copy}}. \tag{22}$$

Let $\hat{O} = f(\hat{\boldsymbol{S}})$ be a general operator acting on $\mathcal{H}_{\mathrm{spin}}$. We associate to $\hat{O}$ the Majorana operator $\hat{\mathcal{O}} = f(\hat{\boldsymbol{\mathcal{S}}})$, acting on $\mathcal{H}_{\mathrm{Maj}}$. We write, using Eq. (20) and Eq. (22)

$$\langle \hat{\mathcal{O}} \rangle_{\hat{\mathcal{H}}_{\mathrm{Maj}}} = \frac{\mathrm{Tr}_{\mathrm{Maj}}\,(\hat{\mathcal{O}}\, e^{-\beta \hat{\mathcal{H}}_{\mathrm{Maj}}})}{\mathrm{Tr}_{\mathrm{Maj}}\, e^{-\beta \hat{\mathcal{H}}_{\mathrm{Maj}}}} = \frac{\mathrm{Tr}_{\mathrm{spin}}\,(\hat{O}\, e^{-\beta \hat{H}})}{\mathrm{Tr}_{\mathrm{spin}}\, e^{-\beta \hat{H}}} = \langle \hat{O} \rangle_{\hat{H}}, \tag{23}$$

which shows that the thermal expectation values of spin-$\frac{1}{2}$ operators can be computed with the Majorana Hamiltonian $\hat{\mathcal{H}}_{\mathrm{Maj}}$.

### 2.2.4    Interacting-level gauge choice in the Majorana Hamiltonian

Any non-constant choice for the copy Hamiltonian $\hat{\mathcal{K}}$ makes the Majorana Hamiltonian $\hat{\mathcal{H}}_{\mathrm{Maj}}$ not $\mathbb{Z}_2$ gauge invariant in the sense of Sec. 2.1.4. Nevertheless, any choice of $\hat{\mathcal{K}}$ is as good as any other for the purpose of computing physical observables, i.e. quantities that only depend on the original spin-$\frac{1}{2}$ operators, as shown in Sec. 2.2.3. This means that using a non-constant $\hat{K}$ is equivalent to choosing a preferred gauge for our calculations, and we refer to this as the interacting-level gauge choice. If no interacting-level gauge choice is made, a gauge choice must be made in the non-interacting Hamiltonian, which we refer to as non-interacting-level gauge choice. See Sec. 2.3.2 for a further discussion of the two levels at which a gauge choice must be made.

The Majorana Hamiltonian gauge invariance is equivalent to the "copy symmetry" of the Hamiltonian with respect to the various copies of the physical spin Hilbert space. An interacting-level gauge choice therefore removes the exponential energy degeneracy between the different copies.

## 2.3    $XYZ$ model and mean-field theory

In this section, we introduce the standard Hartree-Fock mean-field theory for the Majorana-mapped spin-$\frac{1}{2}$ $XYZ$ Hamiltonian. As our main goal is to study systems in which the symmetries of the Hamiltonian are not broken, we focus on the so-called "color-preserving" mean-field theory. For a non-interacting-level gauge choice, which is the case we typically consider in this work, there is a gauge-redundancy of mean-field solutions.

### 2.3.1 $XYZ$ **Hamiltonian and Majorana mapping**

From this point on, we focus our attention to the $XYZ$ spin-$\frac{1}{2}$ Hamiltonian

$$\hat{H} = \frac{1}{2} \sum_{j,l,\alpha} J_{jl}^{\alpha} \, \hat{S}_j^{\alpha} \, \hat{S}_l^{\alpha}, \tag{24}$$

where $J_{jj}^{\alpha} = 0$. According to Eqs. (1) and (16), the Hamiltonian $\hat{\mathcal{H}}$ acting on the Majorana Hilbert space $\mathcal{H}_{\mathrm{Maj}}$ is

$$\hat{\mathcal{H}} = -\frac{1}{16} \sum_{j,l,\alpha,\gamma,\delta} |\epsilon_{\alpha\gamma\delta}| \, J_{jl}^{\alpha} \, \hat{\rho}_j^{\gamma} \, \hat{\rho}_j^{\delta} \, \hat{\rho}_l^{\gamma} \, \hat{\rho}_l^{\delta}. \tag{25}$$

### 2.3.2 **Gauge choice at the interacting or non-interacting level**

In this work, we fix copy Hamiltonian $\hat{\mathcal{K}} = 0$. Using the notation of Sec. 2.2.2, we get $\hat{\mathcal{H}}_{\mathrm{Maj}} = \hat{\mathcal{H}}$, which preserves the $\mathbb{Z}_2$ gauge invariance of $\hat{\mathcal{H}}$. With this non-interacting-level gauge choice, the ground state of $\hat{\mathcal{H}}_{\mathrm{Maj}}$ has a degeneracy of $2^N$, the dimension of $\mathcal{H}_{\mathrm{spin}}$.

An alternative non-zero choice for $\hat{\mathcal{K}}$ would be

$$\hat{\mathcal{K}} = \sum_{j,l} K_{jl} \, \hat{\mathcal{J}}_{jl}^z, \tag{26}$$

for $K_{jl} \in \mathbb{R}$, which lifts the gauge degeneracy of the $\hat{\mathcal{H}}$ ground states. In Appendix A we solve the two-site Heisenberg model with the addition of this term. In the two-site model this type of interacting-level gauge choice for $\hat{\mathcal{K}}$ can open a gap between the two copies of the physical system, and this could have advantages for the convergence of the perturbative expansion. An investigation along this direction in a more systematic framework is left for future work.

### 2.3.3 **Hartree-Fock decomposition**

The $XYZ$ Hamiltonian $\hat{\mathcal{H}}$ of Eq. (25) is a fully interacting Majorana Hamiltonian, and it is therefore difficult to treat analytically. For this reason, we consider, as a starting point for our study, the quadratic Hamiltonian obtained with the Hartree-Fock mean-field theory. For $\alpha \neq \gamma$ and $j \neq l$, we write

$$\begin{aligned}
\hat{\rho}_j^{\alpha} \hat{\rho}_j^{\gamma} \hat{\rho}_l^{\alpha} \hat{\rho}_l^{\gamma} \approx &\hat{\rho}_j^{\alpha} \hat{\rho}_j^{\gamma} \langle \hat{\rho}_l^{\alpha} \hat{\rho}_l^{\gamma} \rangle_{\hat{\mathcal{H}}_0} + \hat{\rho}_l^{\alpha} \hat{\rho}_l^{\gamma} \langle \hat{\rho}_j^{\alpha} \hat{\rho}_j^{\gamma} \rangle_{\hat{\mathcal{H}}_0} + \hat{\rho}_j^{\alpha} \hat{\rho}_l^{\gamma} \langle \hat{\rho}_j^{\gamma} \hat{\rho}_l^{\alpha} \rangle_{\hat{\mathcal{H}}_0} + \hat{\rho}_j^{\gamma} \hat{\rho}_l^{\alpha} \langle \hat{\rho}_j^{\alpha} \hat{\rho}_l^{\gamma} \rangle_{\hat{\mathcal{H}}_0} \\
&- \hat{\rho}_j^{\alpha} \hat{\rho}_l^{\alpha} \langle \hat{\rho}_j^{\gamma} \hat{\rho}_l^{\gamma} \rangle_{\hat{\mathcal{H}}_0} - \hat{\rho}_j^{\gamma} \hat{\rho}_l^{\gamma} \langle \hat{\rho}_j^{\alpha} \hat{\rho}_l^{\alpha} \rangle_{\hat{\mathcal{H}}_0} - \langle \hat{\rho}_j^{\alpha} \hat{\rho}_j^{\gamma} \hat{\rho}_l^{\alpha} \hat{\rho}_l^{\gamma} \rangle_{\hat{\mathcal{H}}_0}
\end{aligned} \tag{27}$$

where

$$\hat{\mathcal{H}}_0 = -\frac{i}{2} \sum_{j,l,\alpha,\gamma} A_{jl}^{\alpha\gamma} \, \hat{\rho}_j^{\alpha} \, \hat{\rho}_l^{\gamma} + \frac{1}{16} \sum_{j,l,\alpha,\gamma,\delta} |\epsilon_{\alpha\gamma\delta}| \, J_{jl}^{\delta} \, \langle \hat{\rho}_j^{\alpha} \hat{\rho}_j^{\gamma} \hat{\rho}_l^{\alpha} \hat{\rho}_l^{\gamma} \rangle_{\hat{\mathcal{H}}_0} \tag{28}$$

and $A$ is real and obtained self-consistently. Different self-consistent solutions exist. The solutions can be sought in sets of $A_{jl}^{\alpha\gamma}$ (called ansätze) with some constraints. Two examples are given in the following subsections.

This mean-field decoupling typically breaks the gauge invariance of $\hat{\mathcal{H}}$, which implies that several sets of $A_{jl}^{\alpha\gamma}$ are equivalent up to gauge transformations. This is discussed in Sec. 2.3.6.

While our discussion of the Hartree-Fock decomposition of $\hat{\mathcal{H}}_{\mathrm{Maj}}$ focuses on the case $\hat{\mathcal{K}} = 0$, it is straightforward to generalize it by applying the mean-field decomposition to $\hat{\mathcal{K}}$ as well.

The non-interacting Hamiltonian $\hat{\mathcal{H}}_0$ is solved by a numerical diagonalization of the matrix $A$, which is of size $6N \times 6N$. For the translation invariant mean-field ansätze we consider in this work, the $6N \times 6N$ matrix is Fourier transformed into $N/m$ matrices of size $6m \times 6m$, with $m$ the number of sites in the mean-field unit cell. We give some further details about the diagonalization in Appendix B.

### 2.3.4   Curie-Weiss mean-field theory

The choice

$$\langle \hat{\rho}_l^\alpha \ \hat{\rho}_l^\gamma \rangle_{\hat{\mathcal{H}}_0} \neq 0, \qquad \langle \hat{\rho}_j^\alpha \ \hat{\rho}_l^\gamma \rangle_{\hat{\mathcal{H}}_0} = \langle \hat{\rho}_j^\alpha \ \hat{\rho}_l^\alpha \rangle_{\hat{\mathcal{H}}_0} = 0 \tag{29}$$

is equivalent to the standard Curie-Weiss mean-field theory, which, written in a more familiar form, is just $\hat{S}_j^\alpha \hat{S}_l^\alpha \approx \hat{S}_j^\alpha \langle \hat{S}_l^\alpha \rangle + \langle \hat{S}_j^\alpha \rangle \hat{S}_l^\alpha - \langle \hat{S}_j^\alpha \rangle \langle \hat{S}_l^\alpha \rangle$. This mean-field theory can be applied only if the magnetization is not zero.

### 2.3.5   Color-preserving mean-field theory

In this work, we are interested in studying states that do not break the Hamiltonian symmetries, like spin liquids or systems at high-enough temperature. For the $XYZ$ model, the global spin rotations of $\pi$ around axes $x$, $y$ and $z$ are symmetries. States that respect these symmetries cannot have a non-zero magnetization. Since the expectation value $\langle \hat{\rho}_j^\alpha \hat{\rho}_j^\gamma \rangle_{\hat{\mathcal{H}}_0}$ for $\alpha \neq \gamma$ is related to the local magnetization on site $j$, we impose it to be zero. By the same way, for $\alpha \neq \gamma$ and $j \neq l$, $\langle \hat{\rho}_j^\alpha \hat{\rho}_l^\gamma \rangle_{\hat{\mathcal{H}}_0} \neq 0$ is not compatible with the global spin-rotation symmetries. Accordingly, we also set it to zero.

We can then write the Hartree-Fock Hamiltonian, Eq. (28), as

$$\hat{\mathcal{H}}_0 = \frac{i}{2} \sum_{j,l,\alpha} A_{jl}^{\alpha\alpha} \ \hat{\rho}_j^\alpha \ \hat{\rho}_l^\alpha + \frac{1}{16} \sum_{j,l,\alpha,\gamma,\delta} |\epsilon_{\alpha\gamma\delta}| \ J_{jl}^\delta \ \langle -i\hat{\rho}_j^\alpha \hat{\rho}_l^\alpha \rangle_{\hat{\mathcal{H}}_0} \ \langle -i\hat{\rho}_j^\gamma \hat{\rho}_l^\gamma \rangle_{\hat{\mathcal{H}}_0} \tag{30}$$

with the self-consistent equations

$$A_{jl}^{\alpha\alpha} = \frac{1}{4} \sum_{\gamma,\delta} |\epsilon_{\alpha\gamma\delta}| \ J_{jl}^\gamma \ \langle -i\hat{\rho}_j^\delta \hat{\rho}_l^\delta \rangle_{\hat{\mathcal{H}}_0} . \tag{31}$$

Inspired by the Feynman-diagram language of Sec. 2.4, we call this symmetric mean-field theory "color-preserving".

### 2.3.6   $\mathbb{Z}_2$ gauge transformation and non-interacting-level gauge choice

Under a gauge transformation, $\hat{\rho}_j^\alpha \mapsto \sigma_j \ \hat{\rho}_j^\alpha$, the mean field Hamiltonian $\hat{\mathcal{H}}_0$, defined in Eq. (30), transforms as

$$\hat{\mathcal{H}}_0 \mapsto \frac{i}{2} \sum_{j,l,\alpha} \sigma_j \ \sigma_l \ A_{jl}^{\alpha\alpha} \ \hat{\rho}_j^\alpha \ \hat{\rho}_l^\alpha + \frac{1}{8} \sum_{j,l,\alpha,\gamma,\delta} |\epsilon_{\alpha\gamma\delta}| \ J_{jl}^\alpha \ \langle -i\hat{\rho}_j^\alpha \hat{\rho}_l^\alpha \rangle_{\hat{\mathcal{H}}_0} \ \langle -i\hat{\rho}_j^\gamma \hat{\rho}_l^\gamma \rangle_{\hat{\mathcal{H}}_0} . \tag{32}$$

As we have made a non-interacting-level gauge choice by setting $\hat{\mathcal{K}} = 0$, the interacting Hamiltonian $\hat{\mathcal{H}}$ is gauge-invariant. In addition, the physical spin observables are also gauge invariant. This means that a choice of an ansatz $A$ is equivalent to any other gauge-related ansatz: $A_{jl}^{\alpha\alpha} \mapsto \sigma_j \sigma_l A_{jl}^{\alpha\alpha}$.

In a basis diagonalizing the complete set of commuting observables $\hat{\mathcal{J}}_{jk}^z$ and $\hat{\mathcal{S}}_j^z$ given in Sec. 2.1.6, each basis state has a copy index consisting of the $2^N$ possibilities for the eigenvalues of the $\hat{\mathcal{J}}_{jk}^z$ operators, and it is shared by $2^{2N}$ basis states. A gauge transformation permutes the copy indices of these basis states, but does not modify the physical

observables. Thus, we can group gauge-related ansätze into equivalence classes, distinguishable by some physical (gauge-invariant) observable. For any couple of sites $(j,l)$, two equivalent ansätze $A$ and $A'$ verify $A_{jl}^{\alpha\alpha} = \pm A_{jl}^{\alpha\alpha\prime}$. But two ansätze verifying this condition can as well belong to different equivalence classes.

We illustrate this on the example of a linear chain with $J_{jl}^{\alpha} = 0$ on non-neighboring sites. Let $A$ and $A'$ verifying $A_{jl}^{\alpha\alpha} = \pm A_{jl}^{\alpha\alpha\prime}$. On an open chain with sites indexed by $j = 0$ to $2N - 1$, we can successively change the gauge on sites with increasing indices to get $A$ from $A'$. Note that only $2N - 1$ signs have to be chosen, as there are only $2N - 1$ links. But on a closed chain where sites $0$ and $2N$ are identified, there are $2N$ links but we cannot use the same procedure, as each local gauge change on a site changes the sign of $A_{jl}^{\alpha\alpha}$ on two links: the sign of the product $\prod_{j=0}^{2N-1} A_{jj+1}^{\alpha\alpha}$ is gauge invariant. On a more complex lattice, this is true for any closed loop of sites. These signs are called fluxes and allow to classify the ansätze with the same modulus of $A$ into equivalence classes.

### 2.3.7 Lattice symmetries

We already have imposed some spin-rotational symmetry on the ansatz and now want to impose the lattice symmetries, which are useful, for instance, when we look for spin liquid states. This means that the physical observables have to be invariant under the action of the lattice symmetry group $G_{\mathcal{L}}$. As $(A_{jl}^{\alpha\alpha})^2$ is gauge invariant, it has to be conserved by $G_{\mathcal{L}}$, implying that all symmetry-related links have the same $|A_{jl}^{\alpha\alpha}|$. Once the moduli chosen under this constraint, it remains to chose the signs, or equivalently up to gauge transformations, the fluxes. To determine all the flux patterns compatible with a lattice or in other words, all the equivalence classes, we use the projective symmetry groups (PSG). This formalism was first developed in the context of Schwinger boson and Abrikosov fermion mean-field theories [44–47], but it has also been applied to mean-field Majorana Hamiltonians [48–50].

### 2.3.8 Shifted-action formalism

With the goal of calculating corrections to mean-field theory, we use the formalism of Ref. [51] to build a shifted Majorana Hamiltonian $\hat{\mathcal{H}}_{\text{Maj}}(\xi)$ depending on a complex parameter $\xi$ that interpolates between $\hat{\mathcal{H}}_0$ and $\hat{\mathcal{H}}_{\text{Maj}}$

$$\hat{\mathcal{H}}_{\text{Maj}}(\xi) = (1 - \xi)\,\hat{\mathcal{H}}_0 + \xi\,\hat{\mathcal{H}}_{\text{Maj}}. \tag{33}$$

When evaluated at zero coupling constant $\xi = 0$, this $\xi$-dependent Hamiltonian is the non-interacting Hamiltonian, while it gives back $\hat{\mathcal{H}}_{\text{Maj}}$ for $\xi = 1$:

$$\hat{\mathcal{H}}_{\text{Maj}}(\xi = 0) = \hat{\mathcal{H}}_0, \qquad \hat{\mathcal{H}}_{\text{Maj}}(\xi = 1) = \hat{\mathcal{H}}_{\text{Maj}}. \tag{34}$$

The perturbative expansion corresponds to expanding physical observables in powers of $\xi$. To this end, we define

$$\hat{\mathcal{H}}_{\text{int}} = \hat{\mathcal{H}}_{\text{Maj}} - \hat{\mathcal{H}}_0, \tag{35}$$

such that

$$\hat{\mathcal{H}}_{\text{Maj}}(\xi) = \hat{\mathcal{H}}_0 + \xi\,\hat{\mathcal{H}}_{\text{int}}. \tag{36}$$

The physical value for the coupling constant is $\xi = 1$, and all other values of $\xi$ are not related to the physical system described by the original spin-$\frac{1}{2}$ $XYZ$ Hamiltonian $\hat{H}$. However, for the purposes of understanding the analytic behavior of the perturbative expansion, it is important to study the properties of the relevant quantities for $\xi \neq 1$.

### 2.3.9 Non-perturbative expectation value of color-preserving mean-field parameters

We consider here the non-interacting-level gauge choice $\hat{\mathcal{K}} = 0$. As the operator $-i\hat{\rho}_j^\alpha \, \hat{\rho}_l^\alpha$ is not gauge invariant for $j \neq l$, its thermal expectation value with the $\mathbb{Z}_2$-gauge-invariant interacting Majorana Hamiltonian $\hat{\mathcal{H}}$ is zero. Reciprocally, as the mean-field Hamiltonian $\hat{\mathcal{H}}_0$ is not gauge invariant, the mean-field parameters are typically non-zero

$$\langle -i\hat{\rho}_j^\alpha \, \hat{\rho}_l^\alpha \rangle_{\hat{\mathcal{H}}} = 0, \qquad \langle -i\hat{\rho}_j^\alpha \, \hat{\rho}_l^\alpha \rangle_{\hat{\mathcal{H}}_0} \neq 0. \tag{37}$$

Using the shifted-action formalism of Sec. 2.3.8, this shows that the average value of $i\hat{\rho}_j^\alpha \, \hat{\rho}_l^\alpha$ is zero at $\xi = 1$, and not necessarily zero otherwise. This is due to the fact that $\hat{\mathcal{H}}_{\text{Maj}}(\xi)$ is gauge-invariant only for $\xi = 1$. This means that the non-perturbative values for the color-preserving mean-field parameters are zero.

It is possible to avoid this rather counterintuitive situation using $\hat{\mathcal{H}}_{\text{Maj}}(\xi)$ with an interacting gauge choice $\hat{\mathcal{K}} \neq 0$, see Sec. 2.2.4. While we leave the exploration of this possibility as future work, as already mentioned we detail the solution of the two-site Heisenberg model with an interacting-level gauge choice in App. A.

## 2.4 Feynman diagrams for the expansion around the color-preserving mean-field theory

The present discussion pertains to the systematic calculation of corrections to the color-preserving mean-field theory. The Feynman rules for diagram construction and the fermionic sign are derived. In this section, we present the Feynman diagrams up to second order for the spin susceptibility as an application. The numerical evaluation of these diagrams can be found in the subsequent Results section, see Sec. 3.

### 2.4.1 Matsubara spin susceptibility

The Matsubara (or imaginary-time) spin susceptibility is expressed as follows

$$\chi_{jl}^{\alpha\gamma}(\tau) = \left\langle T_\tau[\hat{S}_j^\alpha(\tau)\,\hat{S}_l^\gamma(0)]\right\rangle_{\hat{H}} = \left\langle T_\tau[\hat{\mathcal{S}}_j^\alpha(\tau)\,\hat{\mathcal{S}}_l^\gamma(0)]\right\rangle_{\hat{\mathcal{H}}_{\text{Maj}}}, \tag{38}$$

where we have applied Eq. (23) and used the time-ordering operation $T_\tau$. The imaginary-time operators in the Heisenberg picture are defined as

$$\hat{S}_j^\alpha(\tau) = e^{\tau\hat{H}}\,\hat{S}_j^\alpha\,e^{-\tau\hat{H}}, \qquad \hat{\mathcal{S}}_j^\alpha(\tau) = e^{\tau\hat{\mathcal{H}}_{\text{Maj}}}\,\hat{\mathcal{S}}_j^\alpha\,e^{-\tau\hat{\mathcal{H}}_{\text{Maj}}}. \tag{39}$$

From the imaginary-time susceptibility it is possible to obtain, for instance, the static structure factor and, by analytic continuation, the dynamic structure factor as well.

### 2.4.2 Perturbative expansion in the shifted-action formalism

The shifted-action formalism of Sec. 2.3.8 is used to systematically build the perturbative expansion around the mean-field theory. We define a $\xi$-dependent susceptibility as

$$\chi_{jl}^{\alpha\gamma}(\tau;\xi) := \left\langle T_\tau[\hat{\mathcal{S}}_j^\alpha(\tau;\xi)\,\hat{\mathcal{S}}_l^\gamma(0;\xi)]\right\rangle_{\hat{\mathcal{H}}_{\text{Maj}}(\xi)} \tag{40}$$

where we have used the $\xi$-dependent Heisenberg picture

$$\hat{\mathcal{S}}_j^\alpha(\tau;\xi) = e^{\tau\hat{\mathcal{H}}_{\text{Maj}}(\xi)}\,\hat{\mathcal{S}}_j^\alpha\,e^{-\tau\hat{\mathcal{H}}_{\text{Maj}}(\xi)}. \tag{41}$$

By definition, we obtain the physical value $\chi_{jl}^{\alpha\gamma}(\tau)$ of the susceptibility for $\xi = 1$.

We write

$$\chi_{jl}^{\alpha\gamma}(\tau;\xi) = \frac{X_{jl}^{\alpha\gamma}(\tau;\xi)}{\tilde{\mathcal{Z}}_{\mathrm{Maj}}(\xi)}, \tag{42}$$

with

$$X_{jl}^{\alpha\gamma}(\tau;\xi) = \frac{\mathrm{Tr}_{\mathrm{Maj}}\ (T_\tau[\hat{\mathcal{S}}_j^\alpha(\tau;\xi)\,\hat{\mathcal{S}}_l^\gamma(0;\xi)]e^{-\beta\hat{\mathcal{H}}_{\mathrm{Maj}}(\xi)})}{\mathrm{Tr}_{\mathrm{Maj}}\ e^{-\beta\hat{\mathcal{H}}_0}}, \tag{43}$$

$$\tilde{\mathcal{Z}}_{\mathrm{Maj}}(\xi) = \frac{\mathrm{Tr}_{\mathrm{Maj}}\ e^{-\beta\hat{\mathcal{H}}_{\mathrm{Maj}}(\xi)}}{\mathrm{Tr}_{\mathrm{Maj}}\ e^{-\beta\hat{\mathcal{H}}_0}}, \tag{44}$$

and we rewrite these two quantities as the thermal average of two observables under the Hamiltonian $\hat{\mathcal{H}}_0$ [52, 53]

$$X_{jl}^{\alpha\gamma}(\tau;\xi) = \langle\hat{\mathcal{U}}(\xi)T_\tau[\hat{\mathcal{S}}_j^\alpha(\tau;\xi)\,\hat{\mathcal{S}}_l^\gamma(0;\xi)]\rangle_{\hat{\mathcal{H}}_0} \tag{45}$$

$$\tilde{\mathcal{Z}}_{\mathrm{Maj}}(\xi) = \langle\hat{\mathcal{U}}(\xi)\rangle_{\hat{\mathcal{H}}_0}, \tag{46}$$

where

$$\hat{\mathcal{U}}(\xi) = e^{\beta\hat{\mathcal{H}}_0}\ e^{-\beta\hat{\mathcal{H}}_{\mathrm{Maj}}(\xi)} = T_\tau\ e^{-\xi\int_0^\beta d\tau\ \hat{\mathcal{H}}_{\mathrm{int},I}(\tau)}. \tag{47}$$

where $\hat{\mathcal{O}}_I(\tau)$ is the operator $\hat{\mathcal{O}}$ at imaginary time $\tau$ in the interaction picture:

$$\hat{\mathcal{O}}_I(\tau) = e^{\tau\hat{\mathcal{H}}_0}\ \hat{\mathcal{O}}\ e^{-\tau\hat{\mathcal{H}}_0}. \tag{48}$$

We can now formally expand the susceptibility $\chi_{jl}^{\alpha\gamma}(\tau;\xi)$ in powers of $\xi$, as well as $X_{jl}^{\alpha\gamma}(\tau;\xi)$ and $\tilde{\mathcal{Z}}_{\mathrm{Maj}}(\xi)$:

$$\chi_{jl}^{\alpha\gamma}(\tau;\xi) = \sum_{n=0}^\infty \chi_{jl;n}^{\alpha\gamma}(\tau)\,\xi^n, \tag{49}$$

$$X_{jl}^{\alpha\gamma}(\tau;\xi) = \sum_{n=0}^\infty X_{jl;n}^{\alpha\gamma}(\tau)\,\xi^n, \tag{50}$$

$$\tilde{\mathcal{Z}}_{\mathrm{Maj}}(\xi) = \sum_{n=0}^\infty \tilde{\mathcal{Z}}_{\mathrm{Maj};n}\,\xi^n \tag{51}$$

The coefficient of $\xi^n$, $\chi_{jl;n}^{\alpha\gamma}(\tau)$, provides the $n$-th correction to the Majorana mean-field theory. In this section, we only focus on the calculation of the coefficients $\chi_{jl;n}^{\alpha\gamma}(\tau)$ of Eq. (49), while convergence properties are discussed on an example in Sec. 3.2.

In this way, we can write

$$\tilde{\mathcal{Z}}_{\mathrm{Maj};n} = \frac{(-1)^n}{n!}\int_{[0,\beta]^n} d\tau_1\ldots d\tau_n\ \left\langle T_\tau\left[\hat{\mathcal{H}}_{\mathrm{int},I}(\tau_1)\ldots\hat{\mathcal{H}}_{\mathrm{int},I}(\tau_n)\right]\right\rangle_{\hat{\mathcal{H}}_0} \tag{52}$$

$$X_{jl;n}^{\alpha\gamma}(\tau) = \frac{(-1)^n}{n!}\int_{[0,\beta]^n} d\tau_1\ldots d\tau_n\ \left\langle T_\tau\left[\hat{\mathcal{H}}_{\mathrm{int},I}(\tau_1)\ldots\hat{\mathcal{H}}_{\mathrm{int},I}(\tau_n)\,\hat{\mathcal{S}}_{j,I}^\alpha(\tau)\,\hat{\mathcal{S}}_{l,I}^\gamma(0)\right]\right\rangle_{\hat{\mathcal{H}}_0}. \tag{53}$$

In conclusion, the perturbative expansion for the spin susceptibility $\chi$ is obtained from a connected Majorana correlation function with the non-interacting Hamiltonian $\hat{\mathcal{H}}_0$

$$\chi_{jl;n}^{\alpha\gamma}(\tau) = \frac{(-1)^n}{n!}\int_{[0,\beta]^n} d\tau_1\ldots d\tau_n\ \left\langle T_\tau\left[\hat{\mathcal{H}}_{\mathrm{int},I}(\tau_1)\ldots\hat{\mathcal{H}}_{\mathrm{int},I}(\tau_n)\,\hat{\mathcal{S}}_{j,I}^\alpha(\tau)\,\hat{\mathcal{S}}_{l,I}^\gamma(0)\right]\right\rangle_{\hat{\mathcal{H}}_0}^c, \tag{54}$$

where $\langle\cdot\rangle_{\hat{\mathcal{H}}_0}^c$ means a connected non-interacting average.

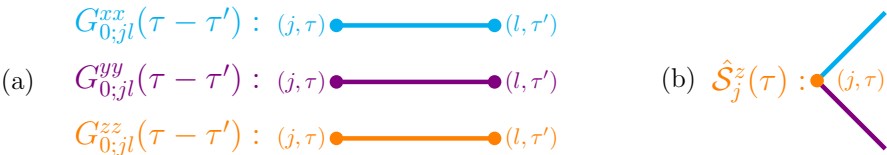

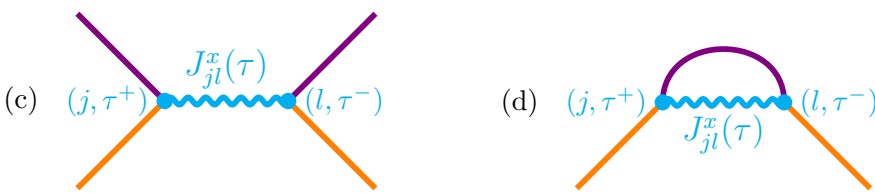

Figure 1: *Building blocks for the Majorana Feynman diagrams.* We associate a color to each of the coordinates $\alpha \in \{x, y, z\}$. (a) the three non-interacting Majorana Green's functions $G_{0;jl}^{\alpha\alpha}(\tau - \tau')$ present in the color-preserving mean-field theory, represented by a colored edge; (b) a Majorana spin operator $\hat{\mathcal{S}}_j^z(\tau)$, represented by a vertex of the $z$-color, and two half-edges of the two other colors; (c) a spin-spin interaction vertex associated to $J_{jl}^x$, represented by a colored wavy line connecting two Majorana-spin-operator vertices of this color; (d) an interaction vertex coming from $\hat{\mathcal{H}}_0$: such vertices eliminate diagrams with Fock insertions in the expansion.

### 2.4.3   Wick's theorem for Majorana fermions

We use the time-ordered version of the Wick's theorem for Majorana fermions presented in Refs. [54–57]

$$(-i)^n \left\langle T_\tau \left[ \hat{\rho}_{j_1,I}^{\alpha_1}(\tau_1) \, \dots \, \hat{\rho}_{j_{2n},I}^{\alpha_{2n}}(\tau_{2n}) \right] \right\rangle_{\hat{\mathcal{H}}_0} = \mathrm{Pf}\, \mathbb{G}_0, \tag{55}$$

where we have introduced the Pfaffian $\mathrm{Pf}\, \mathbb{G}_0$ of the $2n \times 2n$ Green's function matrix $\mathbb{G}_0$, whose components are

$$[\mathbb{G}_0]_{u,v} = G_{0;j_u j_v}^{\alpha_u \alpha_v}(\tau_u - \tau_v) = -i \left\langle T_\tau \left[ \hat{\rho}_{j_u,I}^{\alpha_u}(\tau_u) \, \hat{\rho}_{j_v,I}^{\alpha_v}(\tau_v) \right] \right\rangle_{\hat{\mathcal{H}}_0}. \tag{56}$$

In our case, thanks to the restriction to a color-preserving mean-field theory (see Eq. (30)), the non-interacting Green's function is diagonal in the coordinate index

$$G_{0;jl}^{\alpha\gamma}(\tau - \tau') = \delta^{\alpha\gamma}\, G_{0;jl}^{\alpha\alpha}(\tau - \tau'). \tag{57}$$

### 2.4.4   Feynman rules

Feynman diagrams for the color-preserving mean-field expansion can be constructed through a graphical interpretation of Wick's theorem. A direct consequence of Eq. (57) is that we only have "unicolor" propagators, meaning that it is possible to factorize the whole correlator into a product of three smaller correlators, one for each color. The sign of the Feynman diagram is then just the product of the sign for each color, obtained by a procedure detailed below.

    Fig. 1 gives the graphical representation of the three quartic interaction vertices of $\hat{\mathcal{H}}_{\mathrm{int},I}(\tau)$ and of the quadratic counterterms corresponding to $\hat{\mathcal{H}}_{0,I}(\tau)$, which eliminate all Fock-diagram insertions as a consequence of the self-consistency Eq. (31). With the purpose of clarifying the discussion, we have added a time dependence to the spin interaction

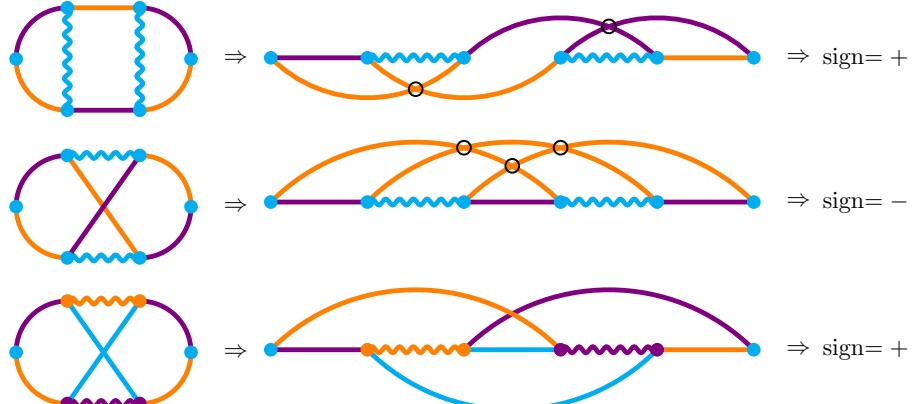

Figure 2: *Examples of the sign determination for Majorana Feynman diagrams.* We first flatten the graph by drawing all the wavy lines on a single line and put edges of the same color either above or below. The final sign is $(-1)^{N_{\text{intersec}}}$, where $N_{\text{intersec}}$ is the total number of intersection of same-color edges (highlighted with an open black circle). Note that the diagrams depicted in the second and the third row correspond to the same "uncolored" Feynman diagram, yet they have opposite signs due to their colors.

$J_{jl}^{\alpha}$; one just needs to set $J_{jl}^{\alpha}(\tau) = J_{jl}^{\alpha}$ at the end of the derivation. We assign three colors to denote the $\{x, y, z\}$ coordinates.

We state here the Feynman rules to draw all diagrams and to determine their weights for the spin susceptibility in the space-time representation. Let $n$ be the order of the expansion for $\chi_{jl}^{\alpha\gamma}(\tau)$. Let $J_{j_1 l_1}^{\alpha_1}(\tau_1) \ldots J_{j_n l_n}^{\alpha_n}(\tau_n)$ be a set of interaction vertices. The Feynman rules are summarized here:

- Draw the two external vertices $\hat{S}_{j,I}^{\alpha}(\tau)$ and $\hat{S}_{l,I}^{\gamma}(0)$ and the interaction vertices $J_{j_k l_k}^{\alpha_k}(\tau_k)$ (see Fig. 1).

- Connect the half-edges of the same color in all the possible ways. Avoid self-interactions (Fock diagrams), since they are eliminated by the contractions with $\hat{\mathcal{H}}_{0,I}$ vertices (see Appendix C). Each edge connecting two vertices corresponds to a non-interacting Green's function $G_{0;jl}^{\alpha\alpha}(\tau_u - \tau_v)$.

- Assign an overall factor of $\frac{(-1)^n}{2^{3n+2}}$ to these diagrams. This prefactor comes from the definition of the expansion in Eq. (54). The former denominator $n!$ is compensated by permutations of the $\hat{\mathcal{H}}_{\text{int},I}(\tau)$ that leads to the same diagram, and replaced by $8^n \times 4 = 2^{3n+2}$ as each $\hat{\mathcal{H}}_{\text{int},I}(\tau)$ brings a $1/8$ prefactor and the spin operators are expressed in Majorana fermions with a prefactor $1/4$.

- Rewrite the diagram in the flatten form as in Fig. 2. Count $N_{\text{intersec}}$ the number of intersections of Green's function lines of the same color. The additional prefactor is $(-1)^{N_{\text{intersec}}}$.

### 2.4.5 First and second-order corrections to the mean-field spin susceptibility

Zeroth and first order Feynman diagrams for the $xx$ spin susceptibility $\chi_{jl}^{xx}(\tau)$ are given in Fig. 3, while the second-order diagrams are presented in Fig. 4. The zeroth order contribution corresponds to the color-preserving mean-field theory, and the first and second

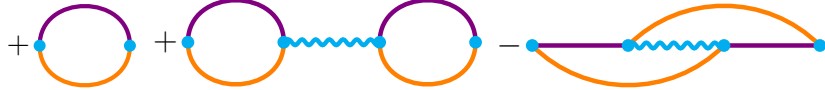

Figure 3: *Mean-field and first-order contributions to the spin susceptibility $\chi_{jl}^{xx}(\tau)$ in the color-preserving mean-field-theory expansion.*

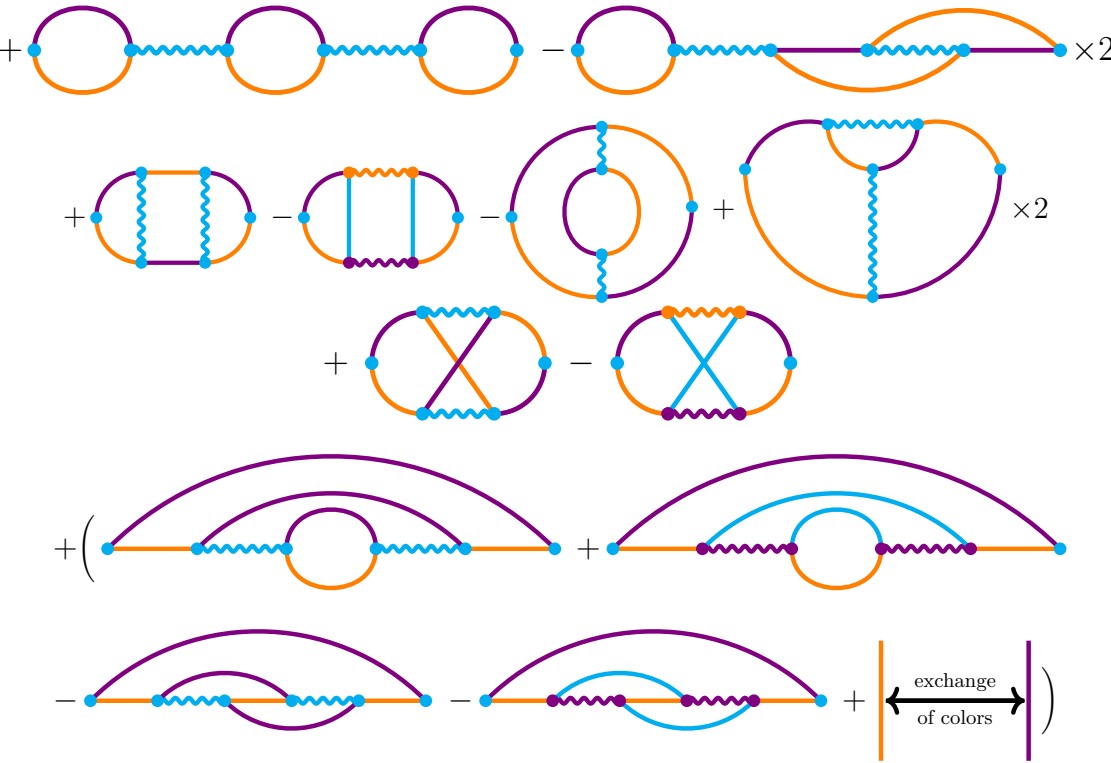

Figure 4: *Second-order contribution to the spin susceptibility $\chi_{jl}^{xx}(\tau)$ for the color-preserving expansion.* A factor of two is added to some diagrams to take into account mirror symmetry. Diagrams in parenthesis are symmetrized with respect to $y$ and $z$ by color exchange. Note that in the "color-symmetric" case discussed in Sec. 3, the diagrams of the third row cancel exactly.

order expressions are corrections to it. We have verified the Feynman diagrams up to second order both symbolically, with Mathematica, and numerically, for small system sizes. In the framework of the color-preserving mean-field expansion, the non-diagonal part of the spin susceptibility is identically zero at all orders.

## 3 Results

In this section, we present numerical results obtained by a direct evaluation of Feynman diagrams up to second order for the Heisenberg model. In this case, the color-preserving mean-field theory introduced in Sec. 2.3.5 becomes color-symmetric, which means that the non-interacting Hamiltonian does not break the spin-rotation symmetry, as we discuss in Sec. 3.1. As a first benchmark of the diagrammatic formalism, in Sec. 3.2 we discuss the properties of the perturbative expansion and evaluate it numerically to high-order in the case of the four-site Heisenberg model, showing that the radius of convergence is finite

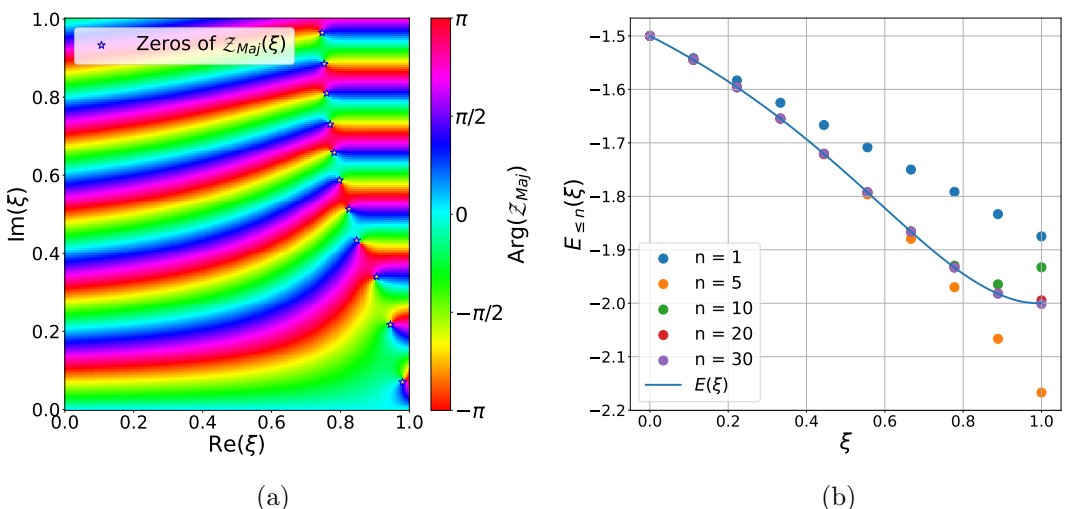

(a)                                                     (b)

Figure 5: *Partition function zeros and convergence of the perturbative expansion for the four-site Heisenberg model with π-flux.* We consider here a temperature $T = 0.05$. (a) Argument of the complex Majorana partition function $\mathcal{Z}_{\text{Maj}}(\xi)$ as function of the complex coupling constant $\xi$. The zeros of $\mathcal{Z}_{\text{Maj}}(\xi)$ correspond to poles in physical observables, and they determine the convergence radius of the perturbative expansion. (b) Convergence of the average-energy partial sums $E_{\leq n}(\xi)$ as function of truncation order $n$ and expansion parameter $\xi$.

at all temperatures and that the diagrammatic series is convergent. In Sec. 3.3 we move on to study the Heisenberg chain, evaluating the first two corrections to the mean-field theory introduced in Ref. [26]. Finally, in Sec. 3.4, we present our second-order results for the square lattice, for the two translation-invariant mean-field theories with zero and $\pi$ flux in the smallest plaquette.

## 3.1 Spin-rotation-symmetric mean-field theory

The numerical results of this work are obtained for the isotropic Heisenberg model on various lattices, defined by the Hamiltonian

$$\hat{H} = \frac{1}{2} \sum_{j,l} J_{jl}\, \hat{\boldsymbol{S}}_j \cdot \hat{\boldsymbol{S}}_l. \tag{58}$$

We consider $J_{jl} = 1$ for nearest neighbors, and $J_{jl} = 0$ otherwise. Our perturbative starting point is the color-preserving mean-field theory of Sec. 2, which, in the case of an isotropic Heisenberg model, becomes color-symmetric [26, 48, 50]

$$-i \, \langle \hat{\rho}_j^x\, \hat{\rho}_l^x \rangle_{\hat{\mathcal{H}}_0} = -i \, \langle \hat{\rho}_j^y\, \hat{\rho}_l^y \rangle_{\hat{\mathcal{H}}_0} = -i \, \langle \hat{\rho}_j^z\, \hat{\rho}_l^z \rangle_{\hat{\mathcal{H}}_0}, \tag{59}$$

where $\hat{\mathcal{H}}_0$ is defined in Eq. (30). Performing a global spin rotation is equivalent, thanks to the discussion of Sec. 2.1.3, to a global rotation of the Majorana vector $\hat{\boldsymbol{\rho}}_j = (\hat{\rho}_j^x, \hat{\rho}_j^y, \hat{\rho}_j^z)^T$, which leaves invariant the non-interacting Hamiltonian $\hat{\mathcal{H}}_0$ in the case of a color-symmetric mean-field theory. This means that all expectation values with the $\hat{\mathcal{H}}_0$ Hamiltonian are also global-spin-rotation symmetric.

## 3.2 Four-site Heisenberg model: test of the convergence properties of the perturbative expansion

In order to understand the convergence properties of the perturbative expansion developed in Sec. 2, we study the Heisenberg model on a single square of sites. The two-site Majorana-mapped Heisenberg model is analytically solvable, and it is treated in Appendix A. We apply exact diagonalization to the Majorana Hamiltonian $\hat{\mathcal{H}}_{\text{Maj}}(\xi)$ (see Sec. 2.3.8), where we chose self-consistent mean-field parameters with a $\pi$-flux in $\hat{\mathcal{H}}_0$, and we study the complex $\xi$ dependence of the partition function

$$\mathcal{Z}_{\text{Maj}}(\xi) = \text{Tr} \, e^{-\beta \hat{\mathcal{H}}_{\text{Maj}}(\xi)}, \tag{60}$$

and of the average energy

$$E(\xi) = \left\langle \sum_{j=1}^{4} \hat{\boldsymbol{S}}_j \cdot \hat{\boldsymbol{S}}_{j+1} \right\rangle_{\hat{\mathcal{H}}_{\text{Maj}}(\xi)} = \sum_{n=0}^{\infty} E_n \, \xi^n, \tag{61}$$

where we identify site 5 with 1. We also define the partial sums $E_{\leq n}(\xi)$ as

$$E_{\leq n}(\xi) = \sum_{k=0}^{n} E_k \, \xi^k. \tag{62}$$

In Fig. 5 we show the analytic structure of the partition function and the convergence properties of the perturbative series for the average energy. As the partition function of a finite-size system is entire, i.e. it has no singularity in the complex plane, its zeros are discrete. They are also away from the real axis for any finite temperature for a finite system, and, at zero temperature, they converge to a branch cut that reaches the real axis, signaling a "phase transition" associated to the restoration of the gauge invariance for $\xi = 1$. The position of the zeros determines the radius of convergence of the perturbative expansion as the partition function appears in the denominator in the observable. The fact that the zeros of the partition function stay at finite distance from the origin even in the zero temperature limit implies then a non-zero convergence radius, a fact that is supported by the convergence of the partial sums. This shows that, at least for the four-site Heisenberg model, the expansion is well-defined at all temperatures, and it has a finite radius of convergence.

The singularity observed at $\xi = 1$ when $T \to 0$ finds its origin in an energy-level crossing in $\hat{\mathcal{H}}_{\text{Maj}}(\xi)$ for $\xi = 1$, which inevitably occurs due to the restauration of the gauge invariance at this point and its associated four-fold ground-state degeneracy. This shows that the zero-temperature convergence radius for a non-interacting-level gauge choice is upper bounded by 1. However, this level-crossing could be avoided by an interacting-level gauge choice, i.e. choosing a non-zero $\hat{\mathcal{K}}$, hence breaking the $\mathbb{Z}_2$ redundancy and the degeneracy (see Appendix A for a demonstration of this idea on the two-site model). This should increase further the radius of convergence of the series.

Contrarily to the $\pi$-flux, the zero-flux ansatz leads to ill-defined series in the limit $T \to 0$, with a zero-temperature zero convergence radius due to a degeneracy of the ground state even at $\xi = 0$.

## 3.3 Heisenberg chain

We consider $2N = 64$ sites with periodic boundary conditions for the spins

$$\hat{H} = \sum_{x=0}^{2N} \hat{\boldsymbol{S}}_x \cdot \hat{\boldsymbol{S}}_{x+1}, \tag{63}$$

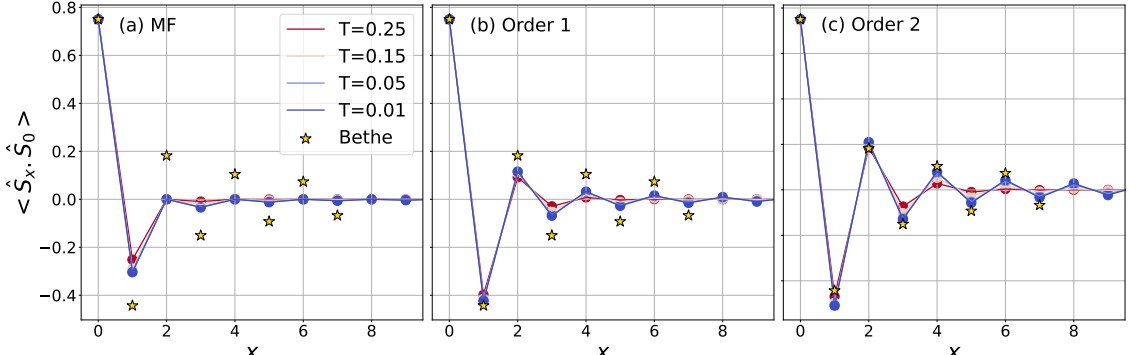

Figure 6: *Spin correlation function for the Heisenberg chain as function of perturbative order.* Results for $2N = 64$ sites with antiperiodic boundary conditions for the Majorana particles ($\pi$-flux). We show the spin correlation function for the color-symmetric mean-field theory (a), at first order (b), and at second order (c). The results are compared with the exact Bethe-ansatz solution for the infinite system at zero temperature [58–60].

where we have used periodic boundary conditions. We use here a $\pi$-flux ansatz that avoids the non-interacting ground-state degeneracy of the zero-flux ansatz. In the thermodynamic limit, both zero-flux and $\pi$-flux ansätze are equivalent, as they correspond to periodic or antiperiodic boundary conditions for the Majorana fermions. The color-symmetric Majorana mean-field solution of this model was introduced in Ref. [26]. The first order was already published in Ref. [27], although in an incorrect form. In this work, we compute corrections up to second order for the spin correlation function

$$C(x) = \sum_{\alpha} \chi_{x0}^{\alpha\alpha}(\tau = 0) = \langle \hat{\boldsymbol{S}}_x \cdot \hat{\boldsymbol{S}}_0 \rangle \,. \tag{64}$$

In Fig. 6 we show the spin correlation function as function of expansion order for $\xi = 1$. The mean-field result is incorrect even at the qualitative level, as the spin correlation function between sites at even distance from each other is zero. The perturbative corrections bring the antiferromagnetic correlations, and they are in good agreement with the exact Bethe-ansatz solution. However, the correlations at large distance are not easy to capture even at second order; the mean-field level exhibits a decay in $1/x^2$ and even if the exponent is reduced by the corrections, the expected decay in $1/x$ is not reached at second order. This could be improved by considering resummed diagrammatic schemes for the spin susceptibility in the spin channel. At zero order, this is related to the Majorana-SO(3) version of the RPA work of Ref. [61].

### 3.4 Square-lattice model

We consider here the square-lattice Heisenberg model

$$\hat{H} = \sum_{x,y} \left( \hat{\boldsymbol{S}}_{x,y} \cdot \hat{\boldsymbol{S}}_{x+1,y} + \hat{\boldsymbol{S}}_{x,y} \cdot \hat{\boldsymbol{S}}_{x,y+1} \right), \tag{65}$$

with periodic boundary conditions. We consider the non-interacting Hamiltonian $\hat{\mathcal{H}}_0$ obtained from a color-symmetric mean-field theory with translational invariance of the physical variables

$$\hat{\mathcal{H}}_0 = iA \sum_{x,y,\alpha} \left[ \zeta_{x,y}^{\boldsymbol{e}_x} \hat{\rho}_{x,y}^{\alpha} \hat{\rho}_{x+1,y}^{\alpha} + \zeta_{x,y}^{\boldsymbol{e}_y} \hat{\rho}_{x,y}^{\alpha} \hat{\rho}_{x,y+1}^{\alpha} \right], \tag{66}$$

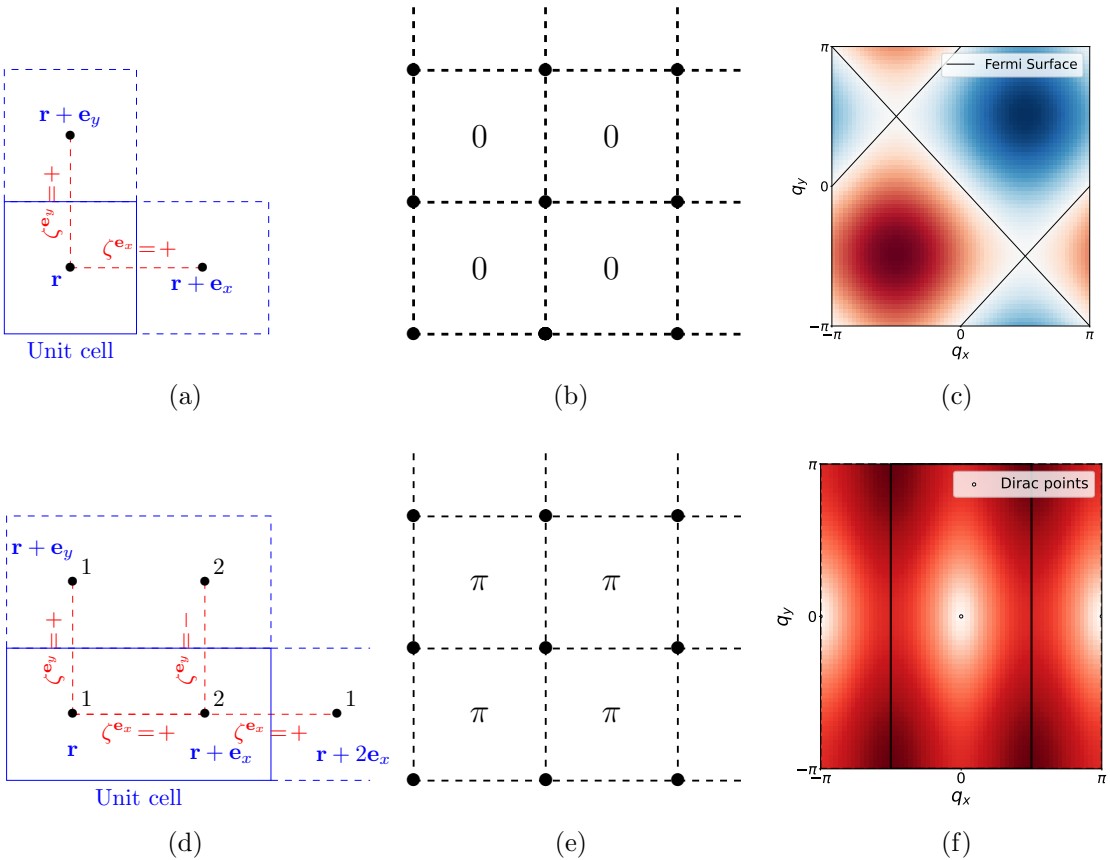

Figure 7: *Uniform zero-flux and $\pi$-flux mean-field ansätze on the square lattice. The first (second) row is about the zero ($\pi$) flux ansatz. (a)(d) Unit cell and choice of $\zeta$ , (b)(e) fluxes on elementary plaquettes, (c)(f) dispersion relation of the unique (zero-flux) or lowest ($\pi$-flux) energy band.*

where we set the constant term to zero, and where $\zeta_{x,y}^{e_x}, \zeta_{x,y}^{e_y}$ are $4N$ variables in $\{-1,1\}$, of which $2N-1$ of them can be fixed by a gauge choice, while the remaining $2N-1$ others determine the fluxes of the elemental squares (not $2N$ as the sum of all fluxes is 0 modulo $2\pi$), and the remaining 2 variables determine the fluxes of a non contractible loop in the $x$ and $y$ directions. The only two choices that lead to a translation-invariant non-interacting spin system on an infinite lattice are the uniform zero flux or the uniform $\pi$ flux, see Fig. 7, and we discuss both choices in the following. Unlike the chain, these two local-flux choices are distinct in the thermodynamic limit as they correspond to different local physical quantities (the flux on a square is such a quantity). On a finite-size lattice, each local-flux choice subdivides into four global gauge-inequivalent $\zeta$ patterns, depending on the boundary conditions for the Majorana fermions: the flux of a non contractible loop in the $x$ or $y$ directions is zero ($\pi$) for (anti-)periodic boundary conditions of the Majorana fermions.

### 3.4.1  Zero-flux expansion

We first consider the zero-flux case, for which a realization in terms of the $\zeta$ variables of Eq. (66) is (see Fig. 7)

$$\zeta_{x,y}^{e_x} = \zeta_{x,y}^{e_y} = 1. \tag{67}$$

We show in Fig. 8 the results of the numerical evaluation of the zero-flux perturbation

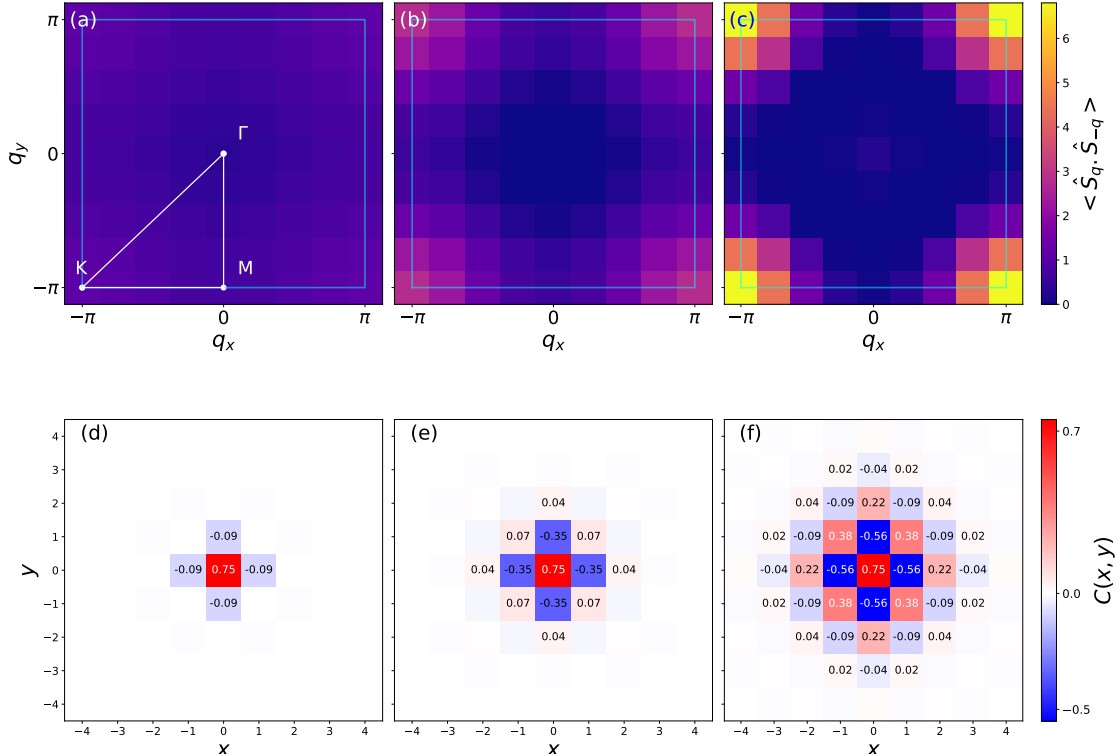

Figure 8: *Spin structure factor and correlation function for the zero-flux perturbative expansion.* We consider a $8 \times 8$ square lattice at $T = 0.25$. We show in the first row the spin structure factor at mean-field level (a), first order (b), and second order (c). In the second row we plot the spin correlation function at mean-field level (d), and up to first (e) and second (f) correction.

theory up to second order for $\xi = 1$ at finite temperature $T = 0.25$ and for a $8 \times 8$ lattice. The zero-flux mean-field theory has a degenerate ground-state, both for periodic and antiperiodic boundary conditions for the Majorana. This creates a formal divergence of the magnetic susceptibility, and, while the effect becomes smaller for increasing lattice size, it forces us to consider an artificially-high finite temperature to avoid it. As noticed for the Heisenberg chain, the perturbative corrections build the missing antiferromagnetic correlations of the mean-field solution, as we can see from both the real space correlation function and the spin structure factor.

### 3.4.2   $\pi$-flux expansion

We now discuss the uniform $\pi$-flux color-symmetric mean-field theory, which can be obtained from the following choice of $\zeta$ in Eq. (66) (see Fig. 7)

$$\zeta_{x,y}^{\boldsymbol{e}_x} = 1, \qquad \zeta_{x,y}^{\boldsymbol{e}_y} = (-1)^x. \tag{68}$$

The spin structure factor and the spin correlation function are presented up to second order for $\xi = 1$ in Fig. 9. The calculations are performed at $T = 0.01$ for a finite $8 \times 8$ system size. As the mean-field ground-state is non-degenerate in the case of uniform $\pi$-flux with antiperiodic boundary conditions, we are able to simulate any value of temperature without unphysical finite-size divergences. Similarly to the zero-flux case, perturbative corrections qualitatively correct the mean-field theory by constructing the antiferromagnetic correlations that were missing at the mean-field level. In Fig. 10 we show that spin

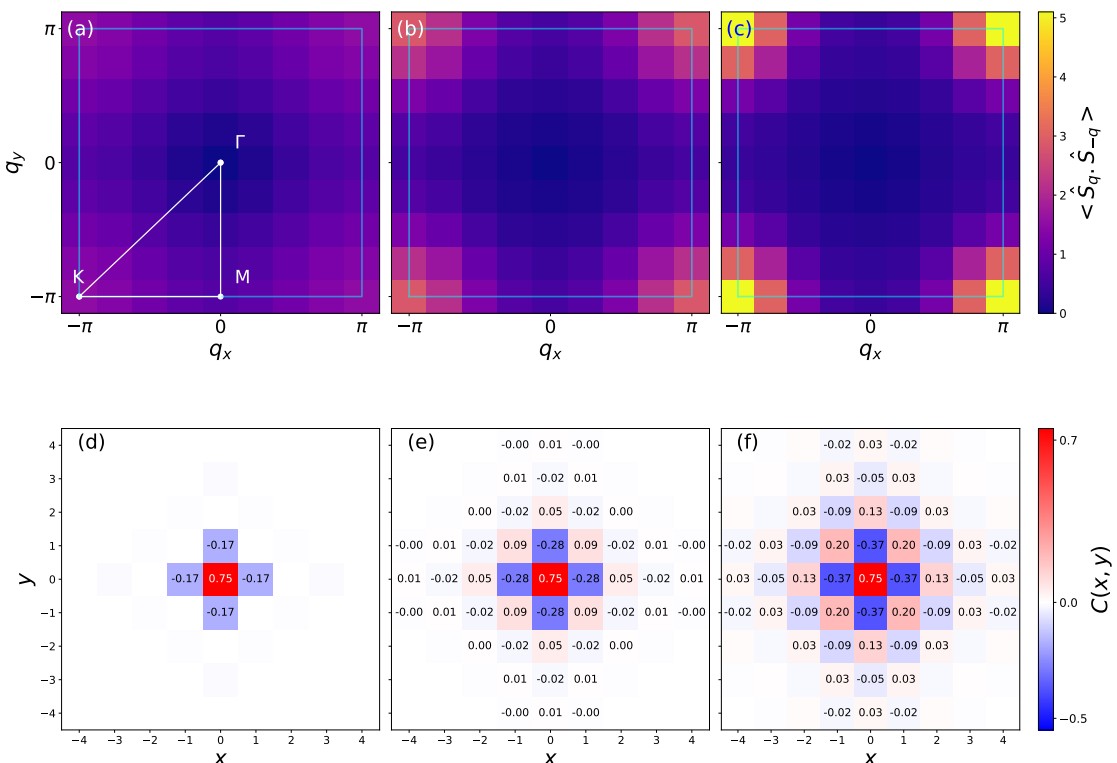

Figure 9: *Spin structure factor and spin correlation function for the $\pi$-flux perturbative expansion.* We consider a $8 \times 8$ square-lattice Heisenberg model at $T = 0.01$. See the caption of Fig. 8 for a description of the inset.

structure factor at the $K$ point increases both as function of inverse temperature and perturbation order, suggesting that the perturbative expansion we introduce in this work is able to progressively retrieve the strong antiferromagnetic correlations of the square-lattice Heisenberg model when pushed to higher orders. This is non trivial as our perturbative starting point, the color-symmetric mean-field theory, is disordered in the thermodynamic limit even at $T = 0$, whereas the $\xi = 1$ point breaks the spin-rotational symmetry.

# 4   Conclusion

In this work we have introduced a systematic diagrammatic formalism for spin-$\frac{1}{2}$ systems based on a unconstrained Majorana fermion mapping. We have first studied the properties of the mapping, with particular attention to the gauge redundancy of the representation. A discussion on how to derive physical spin observables, and the gauge choice at the interacting or non-interacting level, followed. We have then applied the formalism to the $XYZ$ model, where we have introduced a color-preserving mean-field theory that respects the $\pi$-spin-rotation symmetry of the Hamiltonian. We have then derived the Feynman rules to compute corrections to physical spin-$\frac{1}{2}$ observables beyond the color-preserving mean-field theory, and we have explicitly drawn the Feynman diagrams for the spin susceptibility up to second order. Concerning numerical results, we have benchmarked the formalism on the isotropic Heisenberg model. We first analyzed the convergence properties of the diagrammatic expansion for the four-site Heisenberg model, and found a finite radius of convergence at all temperature. We have then extended the mean-field calculation of

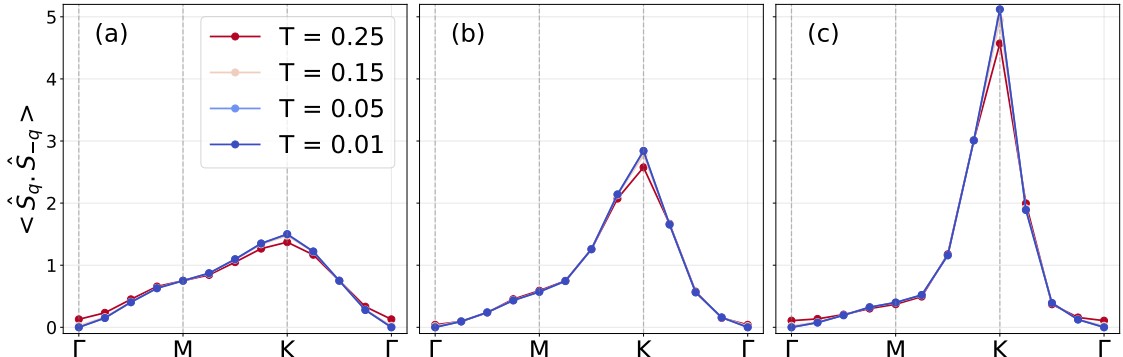

Figure 10: *Spin structure factor along a Brillouin-zone path for the $\pi$-flux expansion.* We show the mean-field (a), first order (b) and second order (c) results for the spin structure factor of the square-lattice Heisenberg model.

Ref. [26] up to second order for the Heisenberg chain, showing that perturbative corrections allows to obtain good agreement with the exact Bethe ansatz solution. We have then shown numerical results for the square-lattice model, for both the uniform zero and $\pi$-flux color-symmetric mean-field theories, and computed the spin correlation function and the spin structure factor. We note that perturbative corrections are able to qualitatively improve the mean-field theory by retrieving antiferromagnetic correlations that are missed at the non-interacting level, which is non-obvious a priori since we expand around a spin-liquid state. These results show that the Majorana diagrammatic formalism for spin-$\frac{1}{2}$ models we present here is a potentially useful theoretical and numerical tool for both ordered and disordered quantum spin systems.

A natural extension of this work consists in computing diagrams to high order in an automatic way with Diagrammatic Monte Carlo algorithm [62–75]. In particular, real-time diagrammatic Monte Carlo techniques [76–82] could be used to calculate the dynamical spin susceptibility, which is a key experimental quantity. As mentioned, we expect an interacting-level gauge choice to further improve the convergence properties of the expansion, and this exploration is left for future work. Finally, in analogy to what is done for in standard many-electron theory, it would be important to understand the effect of diagram resummation for different channels and quantities in the Majorana diagrammatic formalism we introduce here.

Besides technical extensions, further investigations should be led on frustrated spin models with potential SL ground-states, that were the motivation this study. In this article, only unfrustrated lattices have been studied, but the expansion is applicable to any model. However, mean-field Majorana ansatze systematically break the time-reversal symmetry when odd loops are present (due to the real nature of the link parameter $A$, implying a non-zero value for odd products of spin operators). Similarly to the gauge symmetry, the time-reversal symmetry could still be restored at the $\xi = 1$ point.

## Acknowledgements

We thank N. Caci, G. Carleo, A. Grabsch, A. Ralko and A. Tsvelik for insightful discussions. RR acknowledges financial support from SEFRI through Grant No. MB22.00051 (NEQS - Neural Quantum Simulation).

# Appendix

## A    Two-site Heisenberg model and interacting-level gauge choice

### A.1    The Hamiltonian

We consider the Majorana-mapped two-site Heisenberg Hamiltonian $\hat{\mathcal{H}}$, a color-symmetric quadratic Majorana Hamiltonian $\hat{\mathcal{H}}_0$, and a copy Hamiltonian $\hat{\mathcal{K}}$

$$\hat{\mathcal{H}} = -\frac{J}{8}\sum_{\alpha \neq \gamma} \hat{\rho}_1^\alpha \, \hat{\rho}_1^\gamma \, \hat{\rho}_2^\alpha \, \hat{\rho}_2^\gamma, \qquad \hat{\mathcal{H}}_0 = -iA\sum_\alpha \hat{\rho}_1^\alpha \hat{\rho}_2^\alpha + C, \qquad \hat{\mathcal{K}} = K\,\hat{\mathcal{J}}_{12}^z, \qquad (69)$$

where $C, K \in \mathbb{R}$ are constants. This allows to define the $\xi$-dependent shifted Hamiltonian

$$\hat{\mathcal{H}}_{\text{Maj}}(\xi) = \hat{\mathcal{H}}_0 + \xi\left(\hat{\mathcal{H}} - \hat{\mathcal{H}}_0 + \hat{\mathcal{K}}\right). \qquad (70)$$

$\hat{\mathcal{H}}_{\text{Maj}}(\xi = 0)$ is quadratic, while $\hat{\mathcal{H}}_{\text{Maj}}(\xi = 1)$ is a Hamiltonian that can be used to reproduce all the physical spin observables of the two-site Heisenberg model, for every value of $K \in \mathbb{R}$.

### A.2    Energy eigenstates and eigenvalues

We define the operators

$$\hat{d}^\alpha = -i\hat{\rho}_1^\alpha \, \hat{\rho}_2^\alpha, \qquad \hat{d} = \sum_\alpha \hat{d}^\alpha, \qquad (71)$$

which commute with each other, $[\hat{d}^\alpha, \hat{d}^\gamma] = 0$. Using

$$\hat{d}^2 = 3 + \sum_{\alpha \neq \gamma} \hat{d}^\alpha \hat{d}^\gamma, \qquad \hat{d}^3 = 7\hat{d} + 6\hat{d}^x\hat{d}^y\hat{d}^z = 7\hat{d} - 12\,\hat{\mathcal{J}}_{12}^z, \qquad (72)$$

we can write

$$\hat{\mathcal{H}}_0 = A\,\hat{d} + C, \qquad \hat{\mathcal{H}} = -\frac{J}{8}\left(\hat{d}^2 - 3\right), \qquad \hat{\mathcal{K}} = \frac{K}{12}\hat{d}\left(7 - \hat{d}^2\right), \qquad (73)$$

which shows that the three Hamiltonians are a function of $\hat{d}$. We remark a special feature of the two-site model: the non-interacting eigenstates of $\hat{\mathcal{H}}_0$ coincide with the interacting ones for all $\xi$. The possible eigenvalues of $\hat{d}$ are $\pm 3$ (with multiplicity 1) and $\pm 1$ (with multiplicity 3), and they correspond to the following Hamiltonian eigenvalues

$$\mathcal{E}_{\hat{\mathcal{H}}}(\hat{d} = \pm 3) = -\frac{3J}{4}, \qquad \mathcal{E}_{\hat{\mathcal{H}}}(\hat{d} = \pm 1) = \frac{J}{4}, \qquad \mathcal{E}_{\hat{\mathcal{K}}}(\hat{d} = \pm 3) = \mp\frac{K}{2} = \mathcal{E}_{\hat{\mathcal{K}}}(\hat{d} = \mp 1) \quad (74)$$

where $\mathcal{E}_{\hat{O}}$ is the eigenvalue of $\hat{O}$. We classify eigenvalues with the physical spin index, determined by the eigenvalues of the physical total spin $\hat{S}$ and $\hat{S}^z$, and a copy index, corresponding to the eigenvalue of the copy-pair operator $\hat{\mathcal{J}}_{12}^z$. Therefore, we see that the eigenvalue $\hat{d} = \pm 3$ corresponds to the physical spin singlet $S = 0$ with copy index equal to $\mp\frac{1}{2}$, while the eigenvalue $\hat{d} = \pm 1$ corresponds to the triplet $S = 1$ with copy index $\pm\frac{1}{2}$. We now write down the $\xi$-dependent eigenvalues of $\hat{\mathcal{H}}_{\text{Maj}}$

$$\mathcal{E}_{\hat{\mathcal{H}}_{\text{Maj}}(\xi)}(\hat{d} = \pm 3) = \pm 3A + C + \xi\left(-\frac{3J}{4} \mp 3A - C \mp \frac{K}{2}\right) \qquad (75)$$

$$\mathcal{E}_{\hat{\mathcal{H}}_{\text{Maj}}(\xi)}(\hat{d} = \pm 1) = \pm A + C + \xi\left(\frac{J}{4} \mp A - C \pm \frac{K}{2}\right) \qquad (76)$$

### A.3 Color-symmetric mean-field-shifted Hamiltonian: ground state and gap

The color-symmetric mean-field equations are

$$A = -\frac{J}{6}\langle\hat{d}\rangle - \frac{K}{18}\langle\hat{d}\rangle^2, \qquad C = \frac{J}{12}\langle\hat{d}\rangle^2 + \frac{K}{27}\langle\hat{d}\rangle^3. \tag{77}$$

We now consider the low-temperature limit. We are looking for a singlet solution for $J > 0$, and we choose, without loss of generality, a solution such that $\langle\hat{d}\rangle \to 3$ in the zero-temperature limit, which implies

$$A \to -\frac{J+K}{2}, \qquad C \to \frac{3J}{4} + K \tag{78}$$

and therefore

$$\mathcal{E}_{\hat{\mathcal{H}}_{\mathrm{Maj}(\xi)}}(\hat{d}=\pm 3) \to \mp\frac{3}{2}(J+K) + \frac{3J}{4} + K - \frac{1\mp 1}{2}\xi\,(3J+2K) \tag{79}$$

$$\mathcal{E}_{\hat{\mathcal{H}}_{\mathrm{Maj}(\xi)}}(\hat{d}=\pm 1) \to \mp\frac{J+K}{2} + \frac{3J}{4} + K - \frac{1\mp 1}{2}\xi\,(J+2K) \tag{80}$$

Assuming $J + K > 0$, for $\xi = 0$ the ground state has $\hat{d} = 3$. We remark that the $\hat{d} = 3$ and $\hat{d} = 1$ energies are $\xi$ independent. With the choice $K = 0$ (non-interacting-level gauge choice), the ground state is degenerate for $\xi = 1$, which is due to the copy symmetry. For $K < 0$, we have a ground-state level crossing (for the equivalent ansatz choice $\langle\hat{d}\rangle \to -3$, this would have happened for $K > 0$). For $K > 0$, the minimal gap for $\xi \in [0, 1]$ is $\min(K, J)$, which implies that the interacting-level gauge choice $K \geq J$ makes the minimal gap equal to $J$. This means that an interacting-level gauge choice can make the ground-state analytically connected to the physical ground state with a $\xi \in [0, 1]$ path that has a finite gap.

## B Diagonalization of quadratic Majorana Hamiltonians

We consider a generic quadratic Hamiltonian $\hat{\mathcal{H}}_0$ constructed with a set of Majorana operators $\{\hat{\rho}_\mu\}$, indexed by $\mu \in \{1, \ldots, N\}$, such that

$$\hat{\mathcal{H}}_0(\{\hat{\rho}\}) = \sum_{\mu,\nu} i A_{\mu\nu}\hat{\rho}_\mu\hat{\rho}_\nu, \tag{81}$$

where $A$ is a real skew-symmetric matrix of size $N \times N$. We introduce a new set of $N$ Majorana operators $\{\hat{\eta}_\mu\}$ and a new quadratic "doubled" Hamiltonian $\hat{\mathcal{H}}_D$

$$\hat{\mathcal{H}}_D := \hat{\mathcal{H}}_0(\{\hat{\rho}\}) + \hat{\mathcal{H}}_0(\{\hat{\eta}\}) = \sum_{\mu,\nu} i A_{\mu\nu}\big(\hat{\rho}_\mu\hat{\rho}_\nu + \hat{\eta}_\mu\hat{\eta}_\nu\big). \tag{82}$$

The Hamiltonian $\hat{\mathcal{H}}_D$ is the sum of the two commuting terms $\hat{\mathcal{H}}_0(\{\hat{\rho}\})$ and $\hat{\mathcal{H}}_0(\{\hat{\eta}\})$. We can now define a set of $N$ complex fermions for each index $\mu$ such that

$$\hat{f}_\mu = \frac{1}{2}(\hat{\rho}_\mu + i\hat{\eta}_\mu), \quad \hat{f}_\mu^\dagger = \frac{1}{2}(\hat{\rho}_\mu - i\hat{\eta}_\mu), \tag{83}$$

which allows to express the doubled Hamiltonian as a quadratic Hamiltonian of complex fermions which can be easily diagonalized.

For any observable $\hat{\mathcal{O}}$ that is a function of only the $\hat{\rho}$ operators, we can write

$$\langle\hat{\mathcal{O}}\rangle_{\hat{\mathcal{H}}_0} = \frac{\mathrm{Tr}\{e^{-\beta\hat{\mathcal{H}}_0}\,\hat{\mathcal{O}}\}}{\mathrm{Tr}\{e^{-\beta\hat{\mathcal{H}}_0}\}} = \frac{\mathrm{Tr}\{e^{-\beta\hat{\mathcal{H}}_D}\,\hat{\mathcal{O}}\}}{\mathrm{Tr}\{e^{-\beta\hat{\mathcal{H}}_D}\}} = \langle\hat{\mathcal{O}}\rangle_{\hat{\mathcal{H}}_D}, \tag{84}$$

which shows that we can compute expectation values with the $\hat{\mathcal{H}}_D$ Hamiltonian.

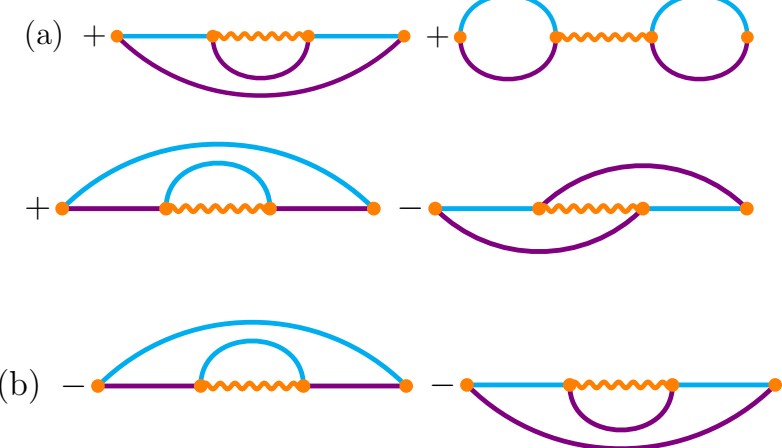

Figure 11: *Elimination of Fock diagrams at first order for the susceptibility.* (a) All Feynman diagrams obtained by the contractions of the $\hat{\mathcal{H}}$ vertices. (b) All Feynman diagrams obtained by the contractions of the $-\hat{\mathcal{H}}_0$ vertices. Disconnected diagrams are canceled by the connected average.

## C  Elimination of Fock diagrams: an example at first order

As mentioned in the Feynamn rules, see Sec. 2.4.4, "self-contracted" Fock diagrams do not contribute to the perturbative expansion around the mean-field theory. This is a consequence of the construction of mean-field counterterms under the self-consistency condition Eq. (31), stating that $A_{jl}^{\alpha\alpha} = \frac{1}{4}\sum_{\gamma,\delta}|\epsilon_{\alpha\gamma\delta}|\,J_{jl}^{\gamma}\,G_{0;ij}^{\alpha\alpha}(0^+)$. As an example showing this fact, we present here the computation of the first-order correction for the spin susceptibility in the direction $z$

$$\chi_{jl;1}^{zz}(\tau) = -\int_0^\beta d\tau_1 \left\langle T_\tau \left[ \hat{\mathcal{H}}_{\text{int},I}(\tau_1)\,\hat{\mathcal{S}}_{j,I}^z(\tau)\,\hat{\mathcal{S}}_{l,I}^z(0) \right] \right\rangle_{\hat{\mathcal{H}}_0}^c. \tag{85}$$

The integrand in the expectation value can be written, without the prefactors, as

$$\left\langle T_\tau \left[ (\hat{\mathcal{H}}_{\text{Maj}}(\tau_1) - \hat{\mathcal{H}}_0(\tau_1))\hat{\rho}_j^x(\tau)\hat{\rho}_j^y(\tau)\hat{\rho}_l^x(0)\hat{\rho}_l^y(0) \right] \right\rangle_{\hat{\mathcal{H}}_0}^c. \tag{86}$$

Only diagrams obtained by contractions with vertices weighted by coupling constant $J^z$, i.e. which have $x$ and $y$ external legs, are contributing. All resulting connected diagrams are presented on Fig. 11, showing diagrammatically that the self-contracted diagrams cancel each other out. This cancellation property remains true for the expansion around the mean-field theory at every perturbative order.

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
