# Peer review of "Majorana Diagrammatics for Quantum Spin-1/2 Models"

_SciPost Physics Core_

## Round 1 · Author Response

We thank both referees for their recomandation of publishing our work and for their comments.

We detail below the changes following the comments and requests of the second referee :

REMARK:
"1) The fact that the mean-field Hamiltonian is a part of the perturbation seems on the first sight to be ignored. Only after some reading is becomes clear that diagrams of Fig. 1d are omitted exactly to take this into account. This is not explained and also the caption of Fig. 1d does not really help. I would recommend to clarify this."

ANSWER:
This is now more apparent in Sec. 2.3.8, where the interaction Hamiltonian (the quartic Hamiltonian minus the mean-field one) is in a dedicated equation.
In Sec. 2.4.4, we now explain that the Fock diagrams are eliminated " as a consequence of the self-consistency Eq. (31)", and in the Feynman rules, we precise that the Fock diagrams are not considered as they "since they are eliminated by the contractions with H_{0,I} vertices (see Appendix C)", refering to a new appendix C detailing the example of the elimination at first order .

REMARK:
"2) It is never mentioned that even the mean-field treatment requires a numerical diagonalization of a large (not exponentially large) matrix. I would recommend to mention this for clarity and explain how the Green's function of the mean-field Hamiltonian are calculated and how the self-consistency is achieved."

ANSWER:
We have added the following paragraph in p.9,
"The non-interacting Hamiltonian Ĥ0 is solved by a numerical diagonalization of the matrix A, which is of size 6N × 6N . For the translation invariant mean-field ansätze we consider in this work, the 6N × 6N matrix is Fourier transformed into N/m matrices of size 6m × 6m, with m the number of sites in the mean-field unit cell. We give some further details about the diagonalization in Appendix B."
that refers to the new Appendix B.

REMARK:
"3) It is not clear that the perturbation parameter ξ is taken to be equal one in both 1D and 2D numerical calculations. I would state this explicitly."

ANSWER:
We have added "for ξ=1" in the section presenting the results for the chain and for the 0-flux and pi_flux ansätze on the square lattice.

REMARK:
"The perturbative approach seems to be useful only for short range physics. There should be a clear physical explanation of this."

ANSWER:
We suppose that by 'short range physics', the referee means 'phases without long-range order' ? The perturbative approach can be useful both for ordered and disordered phases.
On one side, the (similar) bosonic mean-field theory is known to already have both types of phases at the mean-field level (when Bose condensation occurs). On the other side, this is not the case of the fermionic (with complex or Majorana fermions) mean-field theory.
However, this des not prevent the perturbative approach to be useful, as we can study the increase of the correlation length with xi, or add a field that depends on xi and breaks the symmetry at xi=0 but not at xi=1.

---

## Editorial Decision

accepted_in_target_journal